# Steep-slope vertical-transport transistors built from sub-5 nm Thin van der Waals heterostructures

Qiyu Yang[1], Zheng-Dong Luo [1,2] ✉, Huali Duan [3], Xuetao Gan [4] ✉, Dawei Zhang [5,6], Yuewen Li[2], Dongxin Tan[1], Jan Seidel [5,6], Wenchao Chen [3], Yan Liu [1,2] ✉, Yue Hao[1] & Genquan Han[1,2]

Two-dimensional (2D) semiconductor-based vertical-transport field-effect transistors (VTFETs) – in which the current flows perpendicularly to the substrate surface direction – are in the drive to surmount the stringent downscaling constraints faced by the conventional planar FETs. However, low-power device operation with a sub-60 mV/dec subthreshold swing (SS) at room temperature along with an ultra-scaled channel length remains challenging for 2D semiconductor-based VTFETs. Here, we report steep-slope VTFETs that combine a gate-controllable van der Waals heterojunction and a metal-filamentary threshold switch (TS), featuring a vertical transport channel thinner than 5 nm and sub-thermionic turn-on characteristics. The integrated TS-VTFETs were realised with efficient current switching behaviours, exhibiting a current modulation ratio exceeding $1 \times 10^8$ and an average sub-60 mV/ dec SS over 6 decades of drain current. The proposed TS-VTFETs with excellent area- and energy-efficiency could help to tackle the performance degradation-device downscaling dilemma faced by logic transistor technologies.

The continuous technology node shrinking and performance boost of complementary-metal-oxide-semiconductor (CMOS) field-effect transistor (FET) technology have offered significant impetus for the development of the semiconductor integrated circuit industry[1–6]. Despite the physical size downscaling-enabled performance improvement of these transistors, further scaling down along the conventional route has been predicted to be less cost-effective due to the fact that the constituent material and/or device concept are approaching their physics limitations at the nanometre regime[2,4–6]. Especially, the transistor size reduction that limits the device density along with the supply voltage scaling that determines the power consumption have

constituted a daunting challenge for future transistor downscaling[6]. In fact, the device size scaling and the energy-effectiveness increase for future highly scaled nano-transistors have been largely thwarted by the fundamental limits on the device electrostatics and the subthreshold swing (SS) properties, respectively[1]. Therefore, investment in the "next option" for near-future logic transistor technology has urgently promoted the quest for alternative device concepts with innovative materials, structures and working physics[1,2]. Many efforts have been made to enhance the device electrostatics and thus tackle the short-channel effect in ultra-scaled transistors, including FinFETs[7], gate-all-around (GAA) nanosheet FETs[8], vertical-transport FETs (VTFETs)[5],

[1]State Key Discipline Laboratory of Wide Band Gap Semiconductor Technology, School of Microelectronics, Xidian University, Xi'an 710071, China. [2]Hangzhou Institute of Technology, Xidian University, Hangzhou 311200, China. [3]ZJU-UIUC Institute, International Campus, Zhejiang University, Haining 314400, China. [4]Key Laboratory of Light Field Manipulation and Information Acquisition, Ministry of Industry and Information Technology, and Shaanxi Key Laboratory of Optical Information Technology, School of Physical Science and Technology, Northwestern Polytechnical University, Xi'an 710129, China. [5]School of Materials Science and Engineering, UNSW Sydney, Sydney NSW 2052, Australia. [6]ARC Centre of Excellence in Future Low-Energy Electronics Technologies (FLEET), UNSW Sydney, Sydney NSW 2052, Australia. ✉e-mail: zhdluo@xidian.edu.cn; xuetaogan@nwpu.edu.cn; xdliuyan@xidian.edu.cn

beyond-Si semiconductor channel-based transistors[2], etc. Among them, the VTFET—a unique transistor structure in which its source, semiconductor channel and drain components stack vertically to the substrate surface and thus induce the drive current along the perpendicular direction – has been demonstrated as a suitable device architecture to break the device performance-size downscaling dilemma imposed by the conventional planar transistor technology, i.e., lateral-transport FET (LTFET) structure[9–11]. Such a device geometry offers a much-reduced transistor feature by utilizing the vertical construction capability and an ultra-scaled channel length depending on the thickness of the semiconductor layer, which is thus highly promising to achieve an aggressive boost of transistor area-efficiency as well as other desired area-performance device metrics[9–11]. Impressively, Si-based VTFETs have been made possible in commercially-available CMOS technology, which have been regarded as an elegant device strategy to extend the downscaling towards the sub-5 nm CMOS technology node[11]. Furthermore, to maintain the device performance improvement and enrich the device functionalities, various VTFETs device concepts with semiconductor materials beyond Si have been proposed, including but not limited to bulk III-V compound semiconductors[12], organic semiconductors[13], oxide semiconductors[14], and two-dimensional (2D) van der Waals (vdW) materials[6,15–17].

In particular, high-performance 2D vdW heterostructures—which can be created by rationally stacking multiple 2D layers without the lattice-mismatch issue as often experienced in the bulk semiconductor counterparts—have been recently identified as a viable solution for VTFETs[18]. Benefitting from their superior electrostatic control, ultra-scaled 2D semiconductor VTFETs have showcased a well-preserved gate-control ability even at the sub-1-nm transport length limit[15]. Despite the promising merits of 2D semiconductor VTFETs, the evolution of such a device concept towards the ultra-scaled channel length simultaneously with the power consumption cut-down is highly demanding[13,15,18,19]. Although 2D vdW heterostructure-based VTFETs have been identified to have the potential for greatly relaxing the size-reduction constraints of conventional transistor technology, the low-power current switching operation—as set by the fundamental limit on the SS—remains a daunting challenge for energy-efficient VTFETs[20]. Besides, 2D vdW heterostructure-based VTFETs with the channel length downscaling to sub-5 nm generally show moderate transistor performance[19]. Breaking the performance degradation-size downscaling dilemma has thus been identified as a critical obstacle for the development of 2D vdW heterostructure-based VTFETs[19].

Reduction of the SS can efficiently modify the steepness of the conduction state transition slope in a transistor, thus presenting a significantly lowered power consumption through the scaling down of the supply voltage[21]. However, the fundamental thermionic limitation of the SS in a conventional MOSFET is 60 mV/dec at room temperature, and further lowering of the SS beyond thermionic limitation would require alternative device working principles. Meanwhile, a suitable device concept for steep-slope VTFETs featuring both an SS lower than 60 mV/dec and transport channel length less than 5 nm has yet to be explored. Therefore, designing a device architecture with new physical principles becomes the key point to constructing steep-slope VTFETs with ultra-scaled channel length, which would offer a viable approach to tackle the performance enhancement-channel downscaling dilemma for future low-power and area-efficient vertical transistor technology.

In this work, we propose an efficient steep-slope VTFET device architecture formed from the integration of an Ag/TaO$_x$/Ag-based metal filamentary resistance threshold switching (TS) cell and an ultra-scaled MoS$_2$/MoTe$_2$ vdW heterostructure, which is denoted as TS-VTFET. Taking advantage of the abrupt resistance switching of the TS cell along with the gate voltage-modulated conduction transition of the MoS$_2$/MoTe$_2$ heterojunction, unique features such as a sub-5 nm transport channel and a sub-60 mV/dec SS current transition characteristics can be retained in the proposed vdW heterostructure-based TS-VTFET. Specifically, our devices exhibit an average sub-60 mV/dec SS over 6 decades of drain current with a minimum SS (SS$_{min}$) down to ~2.7 mV/dec as well as an excellent current modulation ratio exceeding $1 \times 10^8$ at room temperature. The reliability performance of the TS-VTFETs has been systematically investigated, showing strong voltage-cycling and DC-voltage stressing characteristics that are required for practical transistor applications. Our results provide a promising device concept which could help to address the key downscaling challenges faced by mainstream conventional planar MOSFET technology, forming the basis for the development of future energy- and area-efficient vertical logic transistors.

## Results

### Design strategy and device architecture

For 2D semiconductor-based LTFETs (Fig. 1a), the efficiency of the channel conductance modulation process is limited by the SS (minimum value of 60 mV/dec at room temperature) due to the thermionic injection of carriers over an energy barrier[22]. In recent years, notable innovative steep-slope device concepts have been established which exhibit reduced supply voltage and power consumption by improving the SS beyond the 60 mV/dec limit at room temperature, including but not limited to negative-capacitance FET (NC-FET)[21], tunnelling-FET[6], Dirac-source FET[23], impact ionization FET[24], and phase-FET[25,26]. Especially, unlike other steep-slope transistor architectures, phase-FETs normally comprise disruptive components with different functionalities (that is, the two-terminal phase-transition component integrated in-series with the transistor), where the gate-voltage controlled transistor current transition along with the abrupt resistance switching of the phase-transition component would together lead to an overall sub-60 mV/dec steep-slope switching behaviours[25–28]. Such a heterogeneous integration strategy of disruptive components can be feasibly extended to conventional 2D semiconductor-based VTFETs (Fig. 1b). Therefore, integrating the two-terminal phase-transition device vertically with the 2D vdW heterostructure-based VTFET would naturally maximize the 3D construction capability of the device and result in a compact device architecture to further increase the transistor density on the chip, see Fig. 1c.

Leveraging the electronic phase transition behaviour, such a hybrid architecture combining VTFET and phase-transition component could provide a potential sub-thermionic device concept with both advantages of the excellent area- and energy-efficiency. As shown in Fig. 1c, the proposed VTFET exploits a vdW heterojunction by vertically stacking the unipolar $n$-type (MoS$_2$) and the $p$-terminal dominated ambipolar (MoTe$_2$) semiconductors. Harnessing the merits including minimal interface trap states and large bandgap offset in high-quality MoTe$_2$/MoS$_2$ vdW vertical heterostructures[29], voltage-reconfigurable band alignment can be efficiently realised in such a heterojunction, giving rise to electric field-effect mediated transport behaviours. To further enable the steep-slope current switching in this device architecture, a metal filament formation/rupture-based volatile TS cell (phase-transition component) is proposed to be integrated on the drain terminal of a VTFET without extra area cost, resulting in a small-footprint TS-VTFET structure (Fig. 1c). As the VTFET channel resistance determines the effective drain voltage ($V_d$) drop on the TS and the FET channel, the metal filament formation and rupture behaviours can thus be controlled by the gate voltage ($V_g$), which could consequently give rise to the steep-slope current switch in the proposed TS-VTFETs. Furthermore, benefiting from the high resistance of the off-state TS cell, reduction of the overall off-current would be possible for the TS-VTFET in comparison with the VTFET alone. The working principles of the TS-VTFET will be discussed in detail in the next sections.

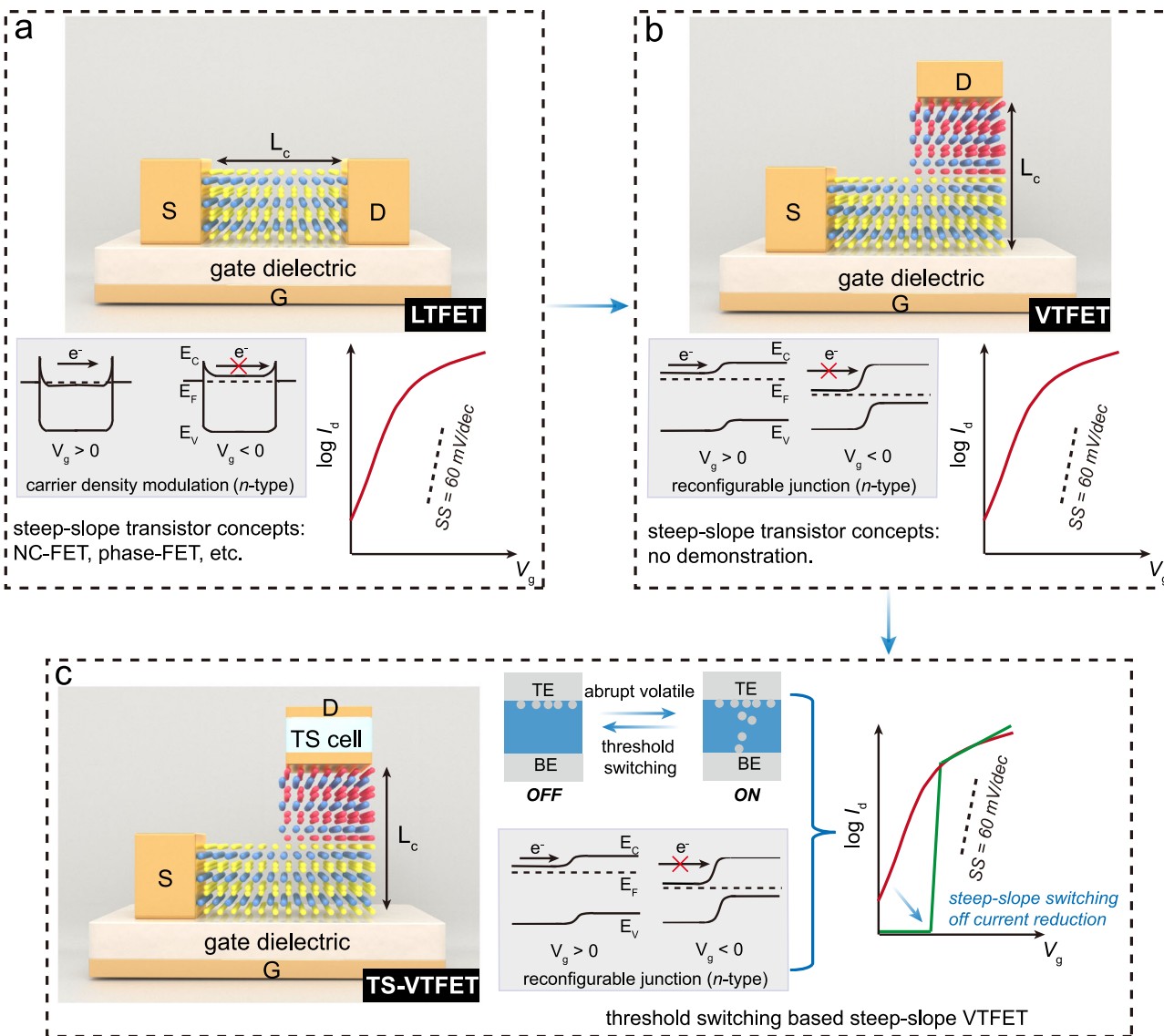

**Fig. 1 | Device architectures and working mechanisms of different 2D material-based field-effect transistors. a** Device architecture of a conventional lateral-transport field-effect transistor (LTFET) where the channel length is denoted as $L_c$. The operation mechanism involving the gate voltage ($V_g$) controllable energy band diagrams at ON and OFF states is depicted along with the typical transfer characteristics. For the energy band diagram, the arrows indicate the electron transport direction. $E_C$, $E_F$, and $E_V$: conduction band energy, Fermi level, and valence band energy, respectively. The drain-source current ($I_d$) switching curves of conventional LTFETs are normally with a subthreshold swing (SS) larger than 60 mV/dec at room temperature. For LTFETs, steep-slope device concepts such as negative-capacitance FET (NC-FET) and phase-FET have been proposed before. **b** Structure of a vdW junction-based vertical-transport field-effect transistor (VTFET). The gate voltage controlled current switching is enabled by the electrostatic modulation of the barrier formed at the junction interface. **c** Threshold switch (TS)-VTFET consists of a VTFET and a TS cell. The abrupt threshold current switching of the TS cell along with the junction current modulation in the VTFET leads to the steep-slope transistor current transition. Moreover, the high resistance performance of the TS cell in series with the VTFET would lead to efficient reduction of the device off-current. TE and BE represents top- and bottom-electrode of the TS cell, respectively.

## Electrical properties of the vdW heterojunction VTFET

Using the standard dry-transfer method with mechanically exfoliated 2D semiconductor flakes (see Supplementary Note 1), we fabricated $MoS_2$ and $MoTe_2$ LTFETs as well as the $MoS_2/MoTe_2$ vdW heterostructure-based VTFET on a $Al_2O_3/HfO_2$ (3 nm/12 nm) dielectric layer, as schematically shown in Fig. 2a. Note that the ultrathin $Al_2O_3$ layer is deposited on the $HfO_2$ for two purposes: (1) suppress the interface defects when in contact with the 2D layers;[30] (2) prevent the leakage favoured crystalline states formed in $HfO_2$[31]. Both $MoS_2$ and $MoTe_2$ layers along with their vertically-stacked heterostructure are investigated by Raman spectroscopy, showing excellent single crystalline quality (Fig. 2a). Among the library of 2D semiconductor-based vdW p-n heterojunctions, such a $MoS_2/MoTe_2$ combination would be more practical to be fabricated from scalable production perspectives since only the alternation of S and Te source is needed with the same Mo source (unlike other 2D heterojunctions like $MoS_2/WSe_2$[32]. The top-view layout of the devices is shown in the optical image (Fig. 2b), where three different devices including a $MoS_2$ LTFET, a $MoTe_2$ LTFET and a $MoS_2/MoTe_2$ VTFET (formed at the overlapping area of the $MoS_2$ and $MoTe_2$ flakes) are demonstrated. The cross-section high-angle annular dark field scanning transmission electron microscopy (HAADF STEM) and high-resolution TEM (HR-TEM) images in Fig. 2c present the layered structure of a typical $MoS_2/MoTe_2$ VTFET created on the $Al_2O_3/HfO_2$ dielectric substrate. Here, an atomically-sharp and ultra-flat interface can be observed between the adjacent $MoS_2$ and $MoTe_2$ layers, which ensures the good performance of the vdW heterostructure-based electronic devices[33]. The uniform distribution of Mo, S, Te, Al and Hf in the $MoTe_2/MoS_2/Al_2O_3/HfO_2/Si$ (bottom

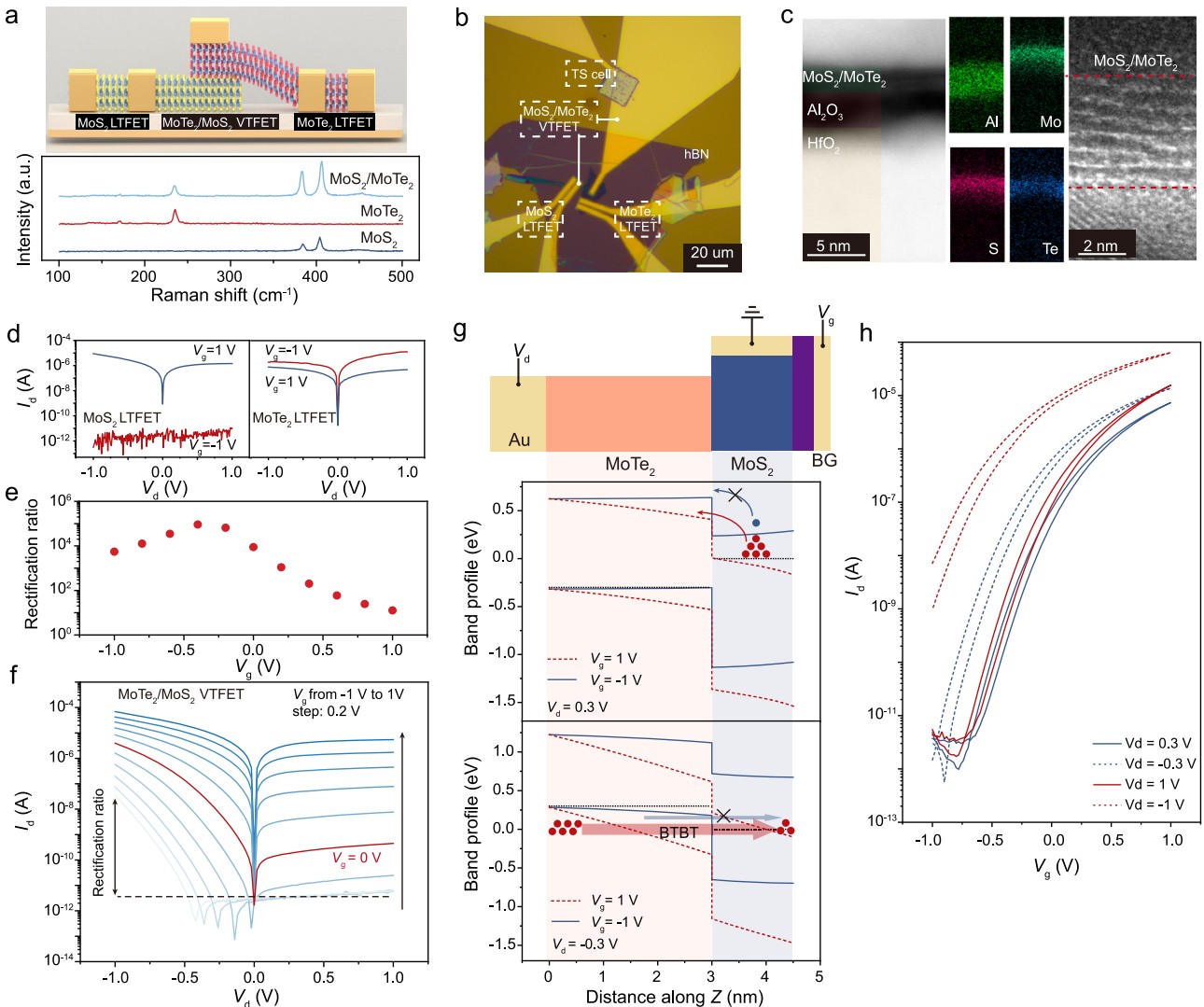

**Fig. 2 | Device structure and performance of 2D material-based LTFETs and VTFETs. a** Schematic diagram showing the $MoS_2$ and $MoTe_2$ LTFETs as well as the $MoS_2/MoTe_2$ vdW heterostructure-based VTFET. Raman spectra of 2D semiconductors are also given. **b** Optical image of the fabricated devices. A vdW hexagonal boron nitride (hBN) layer was placed on top of the devices as a protective layer. **c** Cross-section high-angle annular dark field scanning transmission electron microscopy (HAADF-STEM) and high-resolution TEM (HR-TEM) images of the vertically stacked $MoS_2/MoTe_2$ vdW heterostructure. The distribution of Al, Mo, S and Te is detected using the corresponding energy-dispersive X-ray spectroscopy (EDS) elemental mapping. The red dashed lines indicate the region of the $MoS_2/MoTe_2$ vdW heterostructure. **d** Output curves at different gate voltage values for $MoS_2$ and $MoTe_2$ LTFETs, respectively. $V_d$: drain-source voltage. Output curves (**f**) and rectification ratio (**e**) under different applied gate voltages of the same device as in (**f**). **g** Calculated band profiles of the $MoS_2/MoTe_2$ vdW heterostructure-based VTFET under various $V_d$ and $V_g$. Black dashed line indicates the Fermi energy level, blue and red filled circles represent electrons for different band energy profile cases. The arrows indicate the electron transport direction. **h** Transfer curves of the $MoS_2/MoTe_2$ VTFET measured with different $V_d$.

electrode) has been further confirmed using energy-dispersive X-ray spectroscopy (EDS) mapping, as shown in Fig. 2c. The $MoS_2/MoTe_2$ vdW heterostructure-based VTFET device was also investigated using atomic force microscopy (AFM), showing a thickness of 1.7 nm and 3.2 nm for $MoS_2$ (2 atomic layers) and $MoTe_2$ (4 atomic layers) layer, respectively (see Supplementary Note 2). These results indicate that the transport channel of the fabricated VTFET only has a sub-5 nm channel length, which is highly promising for future ultra-scaled transistor technology[15].

Electrical transport characterizations of all the fabricated $MoS_2$ LTFETs, $MoTe_2$ LTFETs and $MoS_2/MoTe_2$ VTFETs were performed at room temperature. By sweeping the drain-source voltage ($V_d$) under gate voltage with opposite polarities ($V_g = 1$ V and $-1$ V), different output characteristics of $MoS_2$ and $MoTe_2$ LTFETs can be obtained, see Fig. 2d. As a unipolar $n$-type semiconductor, the LTFET with $MoS_2$

channel exhibits high and low conductance state at $V_g = 1$ V and $-1$ V, respectively. In contrast, the LTFET with a $MoTe_2$ channel exhibits almost equally high drain-source current ($I_d$) at $V_g = 1$ V and $-1$ V, indicating the ambipolar nature of the channel semiconductor. Distinct output curves have been observed in the $MoS_2/MoTe_2$ VTFETs, where the metal contact to the bottom $MoS_2$ layer was kept grounded as the source electrode, see Fig. 2e, f. Clear rectification behaviour with asymmetric $I_d$ - $V_d$ output curves can be observed for the $MoS_2/MoTe_2$ heterojunction, which could evolve as a function of the applied gate voltage. Changing the gate voltage from $V_g = -1$ V to $V_g = 1$ V, the rectification ratio drops from nearly $1\times10^5$ to almost 0. This behaviour can be attributed to the gate voltage-induced operation mode switching in the proposed VTFETs.

To gain insight into the working mechanism responsible for the VTFET, we simulated the band profiles along the vertical direction of

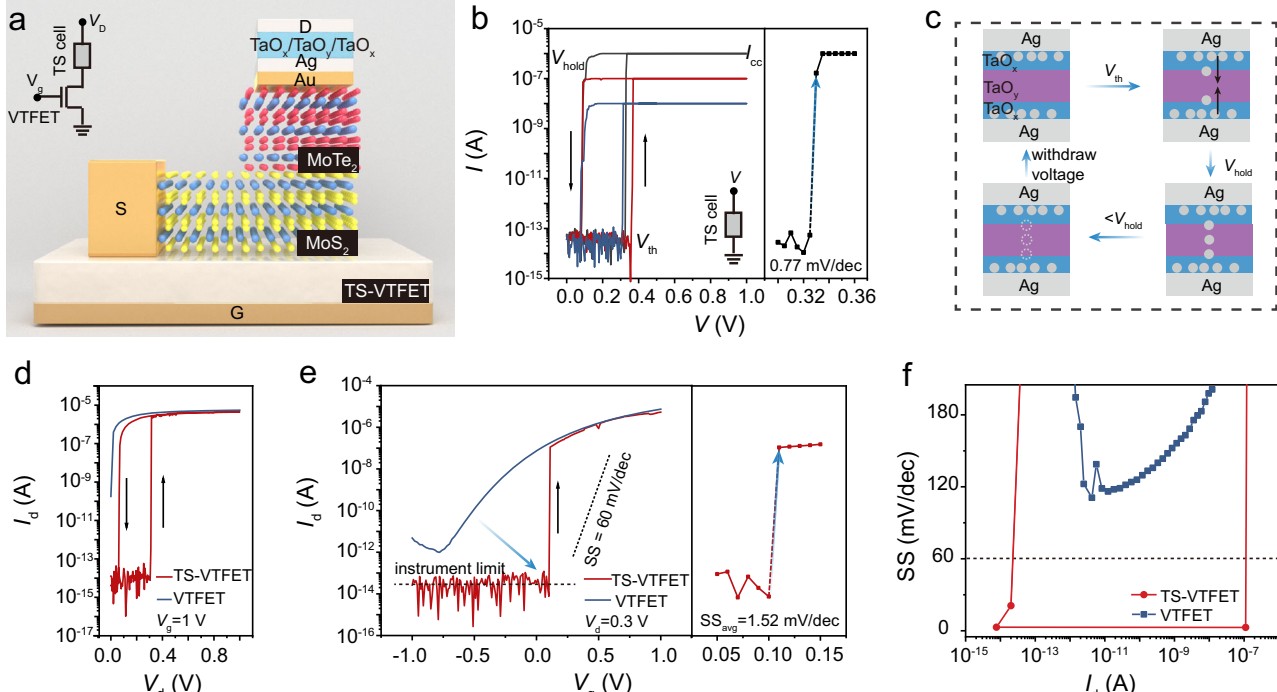

**Fig. 3 | Experimental demonstration of the steep-slope TS-VTFETs. a** Schematic of the $MoS_2$/$MoTe_2$ TS-VTFET, where the source contact is connected to the $MoS_2$ layer. **b** Typical threshold resistive switching of the $TaO_x$/$TaO_y$/$TaO_x$ TS cell with ultra-low leakage current and very steep current switching slope. $V_{hold}$, $I_{cc}$, and $V_{th}$: metal filament holding voltage, compliance current, and threshold voltage triggering the filament formation, respectively. The arrows indicate the current sweeping directions. **c** Schematic drawing of the Ag filament formation and rupture process in the TS cell. **d** Output curves of the VTFET and the TS-VTFET at $V_g = 1$ V.

**e** Transfer characteristics of the VTFET and the TS-VTFET during the forward sweeping process at room temperature. The dashed black line indicates the "cross points" below which the instrument limit induced noisy current contaminates the measured $I_d$. In our case, the instrument limit current cross points occur at or below 30 fA for all the transfer curves measured. **f** Point SS calculation for both VTFET and TS-VTFET as a function of the $I_d$ during forward gate voltage sweeping. The dashed black line indicates the SS = 60 mV/dec limit.

the $MoS_2$/$MoTe_2$ heterojunction under different gate voltages, see Supplementary Note 3 for details. The simulated results show that the gate voltage can efficiently modulate the band profile of the bottom $MoS_2$ layer and control the band alignment of the $MoS_2$/$MoTe_2$ heterojunction as well. Comprehensive band profile of the $MoS_2$/$MoTe_2$ heterojunction-based VTFET under different $V_d$ and $V_g$ are depicted in Fig. 2g. Briefly, corresponding to different polarities of the $V_d$, the proposed VTFET can be operated between a *p-n* junction and a tunnel diode with efficient gate voltage modulation. At $V_d > 0$ V, the $MoS_2$/$MoTe_2$ heterojunction-based VTFET works as a conventional forward biased *p-n* diode, where the current increases with increasing the $V_d$ and the $V_g$ due to the exponential growth of majority carriers in the *n*-type $MoS_2$. The device turns to the high resistance state under negative $V_g$ due to the reason that the $MoS_2$ layer is modulated to the electron depletion state by gate control. On the other hand, the $MoS_2$/$MoTe_2$ heterojunction can work as a reverse biased tunnel diode at $V_d < 0$ V, in which the band-to-band tunnelling (BTBT) current increases with the growth of the applied $V_d$ and the tunnelling onset voltage decreases with higher gate voltage. With decreasing applied $V_g$, the band tilting of the tunnel diode could be weakened, thus resulting in a larger barrier impeding the overall tunnelling current. The above-mentioned working mechanism has been verified with low-temperature $I_d$-$V_d$ characteristics in the VTFETs, where the tunnelling current at negative drain voltage region barely changes but the forward bias current under positive $V_d$ diminishes at 80 K, see Supplementary Note 4. Consistent with the proposed working principles, the $MoS_2$/$MoTe_2$ heterojunction-based VTFET shows distinct transfer characteristics when varying the $V_d$, as shown in Fig. 2h. Similar to a conventional MOSFET, an excellent current switching behaviour with

ON-OFF ratio ($I_{on}/I_{off}$) beyond $1 \times 10^6$ can be achieved, indicating that the $MoS_2$/$MoTe_2$ heterojunction-based VTFET with a sub-5 nm channel transport length could be a promising option for future transistor technology with ultra-scaled channel length. To show the unique gate voltage controllable behaviours of the proposed VTFETs, we have also presented the transfer properties of the $MoS_2$ and $MoTe_2$ LTFETs, see Supplementary Note 5. It is noteworthy that the carrier transport path in the VTFETs consists of the lateral and the vertical directions as discussed in Supplementary Note 6. An ideal VTFET would have a minimal lateral transport channel length to offer the ultra-scaled footprint in the 2D plane, albeit that the lateral conducting line at the bottom linking the vertical channel to the source electrode is unavoidable in VTFET device structure. The lateral transport in the VTFET proceeds by drift-diffusion with the intralayer carrier recombination within each material, while the vertical transport is dominated by the diffusion of majority carriers or the band-to-band tunnelling depending on the applied voltage. As discussed in Supplementary Note 6, the charge transport in VTFET is dominated by the vertical $MoS_2$/$MoTe_2$ *p-n* diode and has no appreciable contribution from the lateral semiconductor channel. This conclusion has been further verified by analysing the electric properties of a $MoS_2$/$MoTe_2$ VTFET with the gate electrode confined within the heterojunction area, see Supplementary Note 7.

**vdW heterostructure-based steep-slope TS-VTFET**

Having demonstrated the basic electrical properties of the $MoS_2$/$MoTe_2$ heterojunction-based VTFETs, we now focus on the steep-slope switching behaviour in the proposed TS-VTFETs (Fig. 3a). Following a previously reported high-performance TS cell design strategy[34], we

fabricated high-quality TS cells using magnetron sputtering, which consist of alternating $TaO_x/TaO_y/TaO_x$ (x < y) layers with Ag as both top and bottom electrodes (see TEM analysis shown in Supplementary Note 8). With such symmetric Ag electrodes, the Ag conductive filament can be triggered to grow at both sides of the TS cell, leading to an excellent volatile bidirectional resistive threshold switching with a very small current switching slope (Fig. 3b). The fabricated TS cell presents reliable threshold switching under pre-set compliance current ($I_{CC}$), showing excellent stability of threshold voltage ($V_{th}$) up to 100 consecutive switching cycles, see Supplementary Note 9. Moreover, the device maintains a high on-off ratio over a wide range of cell areas (see Supplementary Note 9), indicating great uniformity and scalability of the resistive switching medium. We have also discussed the asymmetric electrode configuration effect on the electrical properties of the TS cell, see Supplementary Note 10 for details. As schematically depicted in Fig. 3c, the resistive switching behaviour in the TS cell could be attributed to the formation and rupture of the Ag filament at a proper driving voltage[34]. The Ag filament after the high voltage electroforming process ($V > V_{th}$) could grow from both top- and bottom-Ag electrodes and extend to the $TaO_x/TaO_y/TaO_x$ dielectric medium, which serves as the conducting channel and make the TS cell less insulating. When the applied voltage drops below a certain voltage holding the metal filament formation ($V_{hold}$), the Ag filament becomes unstable and eventually ruptures, thus the conducting channels in the TS cell disappear and the device turns into a highly insulating.

We now demonstrate the TS-VTFET with steep-slope switching properties. As presented in Figs 2b, 3a, the TS cell was directly deposited on the drain terminal of the $MoS_2/MoTe_2$ heterojunction-based VTFET. In this way, we can accurately compare the electrical properties of the VTFET and the TS-VTFET since they share the same $MoS_2/MoTe_2$ heterojunction as the transistor channel. Note that, the $TaO_x/TaO_y/TaO_x$ and Ag layers were deposited at room temperature while the high-quality $Al_2O_3/HfO_2$ dielectric layer were fabricated at 200 °C (see Supplementary Note 11), indicating that the proposed TS-VTFET can be fabricated at low temperature and is thus compatible with the back-end-of-line (BEOL) process. The output curves for both VTFET and TS-VTFET at $V_g = 1$ V are presented in Fig. 3d. As expected, the $MoS_2/MoTe_2$ heterojunction-based TS-VTFET shows a similar $I_d$ - $V_d$ curve with the TS cell, indicating that the transport properties of the TS-VTFET are jointly controlled by the TS phase-transition component as well as the VTFET. For a fixed drain voltage higher than the $V_{th}$ of TS cell ($V_d > V_{th}$), the proposed TS-VTFET can be operated as a transistor with steep switching characteristics, showing a high ON-OFF current switching ratio exceeding $1 \times 10^8$ (Fig. 3e). Compared to the VTFET, an abrupt channel current switching slope can be obtained in the TS-VTFET along with a much-reduced gate voltage modulation range and off-current, see Fig. 3e. Remarkably, in the forward gate bias sweep, the transfer curve of the TS-VTFET at $V_d = 0.3$ V shows excellent sub-threshold characteristics with an average SS of 1.52 mV/dec over 6 orders of magnitude change in the drain current within the $-10^{-14}$ A to $-10^{-7}$ A range. The dashed black line in Fig. 3e indicates the "cross points" below which the drain current of the measured device become similar with the noisy current due to the instrument limit. In our case, the instrument limit current cross points occur at or below 30 fA for all the transfer curves measured on TS-VTFET. The measured off-current from the TS-VTFET exceeds the lower limit of our equipment's capabilities, consequently leading to the generation of random and negative current values. Given such facility limitation, we utilized the lower limit of current detection (30 fA) for the calculation of device performance metrics including SS, ON-OFF ratio, etc., irrespective of the recorded current values. Such a steep-slope switching performance strongly satisfies the Internal Roadmap for Devices and Systems (IRDS) standard for a low-power FET, that is, the average SS is below 60 mV/dec for four decades of channel current. Note, the minimal

current level of our device is determined by the measurement facility limit, which could be even lower than the as-measured value for the actual device and may thus contribute to a steeper current switching with smaller SS[27,34]. Moreover, benefiting from the volatile switching and a low hold voltage of the TS cell, the TS-VTFET can operate with a low hysteresis voltage down to ~60 mV (at $V_d = 1$ V) during gate bias sweeping (Supplementary Note 12), which is important for practical transistor application requiring small hysteresis[35]. Next, statistical data for SS as a function of the output drain current for both VTFET and TS-VTFET is plotted in Fig. 3f. In stark contrast, while the VTFET shows an SS higher than the 60 mV/dec limit at room temperature, the $MoS_2/MoTe_2$ heterojunction-based TS-VTFET presents SS values much below 60 mV/dec over a wide current modulation range and thereby demonstrates better steep-slope switching performance combating the power consumption for transistor operation. The $I_{60}$, a device figure-of-merit that determines the point where the current exhibits a transition from sub-60 mV/dec to super-60 mV/dec, is found to be as high as ~$1 \times 10^{-7}$ A in the presented TS-VTFET. Given the excellent steep-slope device properties, we further compare our device on three important figure-of-merits, i.e., $SS_{min}$, $I_{60}$ and $I_{60}/I_{off}$[36], with previously reported steep-slope transistors. The comparison results clearly indicate excellent performance metrics of the presented TS-VTFET, see detailed discussion in the next section. This underscores its potential as a promising solution for future high-performance steep-slope logic transistors, which is advantageous in terms of miniaturization, device performance, low power consumption, etc.

We now analyse the channel current switching dynamics in the TS-VTFET. At a certain gate voltage, the TS-VTFET can be regarded as a resistor in series with the resistance-variable TS cell (Fig. 4a). As shown in Fig. 4a, the ramped current mode I-V characteristics of the TS cell in series with a load resistor show a clear negative differential resistance (NDR) behaviour with the snapback current switching, indicating the volatile and abrupt formation process of the Ag filament[37]. Correspondingly, by replacing the resistor with the VTFET, it can be predicted that three stages of the current switching process would be identified in the TS-VTFET according to the voltage distribution across the TS cell during gate voltage sweeping, as shown in Fig. 4a, b. Figure 4c schematically illustrates the internal effective voltage distribution and band diagrams of the proposed TS-VTFET at different operation states during forward gate voltage sweeping, indicating that the abrupt current switching mechanism in the TS-VTFET is associated with the gate-voltage controlled TS cell behaviour. At state 1, the VTFET channel is highly insulating with the applied $V_g$ smaller than the threshold gate voltage ($V_{g, th}$), thus the effective drain-source voltage drop on the TS ($V_{TS}$) is much lower than the applied $V_d$ owing to the voltage distribution of the VTFET part ($V_{d'}$). With increasing the gate voltage, partial voltage falling on the TS becomes larger as the resistance of the VTFET reduces. Note, despite the channel resistance of the VTFET decreasing at this stage, the TS-VTFET is still in an insulating condition since the high-resistance TS cell could block the current. With the gate voltage approaching a certain $V_g$ (state 2), the effective drain-source voltage drop on the TS would reach its threshold voltage ($V_{d, th}$) and trigger the formation of Ag filament, thus resulting in the overall drain current surge of the TS-VTFET. Such a process is associated with the NDR behaviour as observed in the TS cell, which could lead to an abruptly increased drain-source voltage drop across the VTFET channel and thereby the steep-slope channel conductance transition. Finally, as shown in stage 3, the TS cell becomes fully conducting and the applied $V_d$ almost totally drops on the VTFET, which gives rise to a normal transfer behaviour similar with that of the VTFET. The above analysis can be further verified by measuring the transfer characteristics of the TS-VTFET with varying the applied drain voltage, as illustrated in Fig. 4b. Obviously, the threshold gate voltage to trigger the channel current switching is dependent on the applied drain voltage of the TS-VTFET,

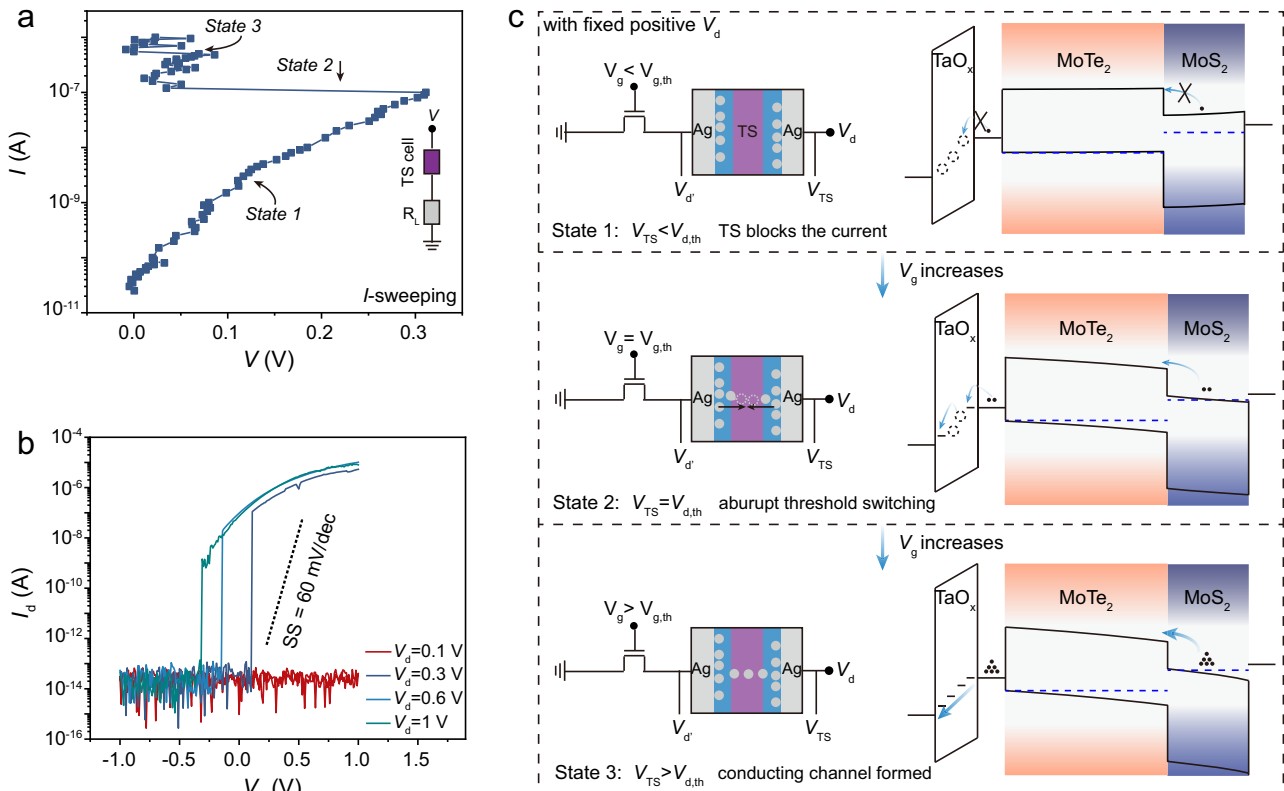

**Fig. 4 | Working mechanism accounting for the steep-slope TS-VTFET.**
**a** Snapback type current switching of the TS cell in series with a 100 kΩ load resistor ($R_L$). By sweeping the current, the TS cell undergoes 3 states: the TS medium is insulating (state 1), Ag metal filament triggered to formation (state 2), and the TS medium enters conducting phase with the formed metal filament (state 3).
**b** Transfer curves of the TS-VTFET under different applied drain voltage values.
**c** Schematic illustration of operation principles for the TS-VTFET associated with the filament formation process in the TS cell. $V_{g, th}$, $V_{d, th}$, $V_{TS}$, $V_d$, and $V_{d'}$: threshold

gate voltage to trigger the TS switching, threshold drain-source voltage to enable the metal filament formation in the TS cell, effective drain voltage on the TS, overall applied drain voltage, and the voltage distribution with the FET channel, respectively. For the band diagrams, the arrows indicate the electron transport direction, horizontal dashed lines represent Fermi level of the 2D semiconductors, horizontal solid lines represent Fermi level of the metals, filled circles denote electrons, and empty circles indicate the broken metal filament conducting path, respectively.

which is due to the balance between the overall drain voltage and the effective potential drop on the VTFET channel during gate sweeping. When applying a $V_d$ less than the $V_{th}$ of the TS cell, the channel conduction state remains insulating during the gate voltage sweeping, since the effective voltage drop on the TS cell constantly remains below its threshold voltage, see the case of $V_d = 0.1$ V shown in Fig. 4b. With increasing the $V_d$ beyond the $V_{d, th}$, the TS cell would be more inclined to be turned into the conducting state as the potential drop on the TS cell gets higher, leading to various transfer properties. Apparently, the change of applied drain voltage can significantly influence the device performance metrics including hysteresis, $I_{60}$ and $V_{th}$, see Supplementary Note 12 for detailed discussions. Overall, this picture clearly indicates that the sub-thermionic switching behaviour of the TS-VTFET is strongly correlated with the TS NDR effect.

## Reliability and performance benchmark of the TS-VTFET
Having discussed the device physics of TS-VTFETs, we now examine the reliability of the proposed device. For a filamentary switching dielectric medium, the high voltage DC stressing would cause a permanent conductive filament formation and thus threshold switching breakdown[37]. As shown in Fig. 5a, the $MoS_2/MoTe_2$ heterojunction-based TS-VTFET can function properly after a 10 s-long high voltage pulse biasing, proving that the demonstrated device is of high capability of long-term service. For the time-dependent DC stressing below the $V_{th}$, the device could work stably without any low voltage

accumulation-induced filament formation, as shown in Fig. 5b. More-over, the TS-VTFET shows great drain-source and gate voltage cycling endurance without any performance degradation after 1000 cycles of ON-OFF switching, see Supplementary Note 13. Overall, these results unambiguously demonstrate that the $MoS_2/MoTe_2$ heterojunction-based TS-VTFETs have excellent reliability.

Finally, we illustrate the performance metrics benchmark of the demonstrated TS-VTFET with state-of-the-art VTFETs, as presented in Fig. 5c and Supplementary Note 14. The $SS_{min}$ in combination with the ON-OFF current ratio at room temperature are highlighted and discussed here as two essential performance metrics for a VTFET. Note that the ON- and OFF-current discussed here refer to the drain current obtained at maximum and minimum gate voltage, respectively. A more detailed comparison of other important device figure-of-merits including current density, $I_{60}$, average SS, etc. can be found in Supplementary Note 15. As can be seen from Fig. 5c, minimum SS value together with the ON-OFF current ratio of our device are compared with representative VTFETs in the literature, including organic semi-conductor (OSC)-based vertical organic FET (VOFET) with ionic liquid gate[13], standard Si VTFET[9], graphene/$MoS_2$ VTFET with ionic liquid gate[38], IGZO/graphene VTFET with $Al_2O_3$ gate[14], VTFET with MXene $Ti_3C_2T_x$/Organic semiconductor PDVT-10[20], $MoS_2$/graphene VTFET[39], $WS_2$/graphene vertical tunnelling transistors[16], black phosphorous (BP)/$MoS_2$ VTFET[29], $MoS_2/MoTe_2$ VTFET on hBN[40] and $MoS_2$/graphene on 300 nm-thick $SiO_2$ dielectric gate[15]. With the capability of

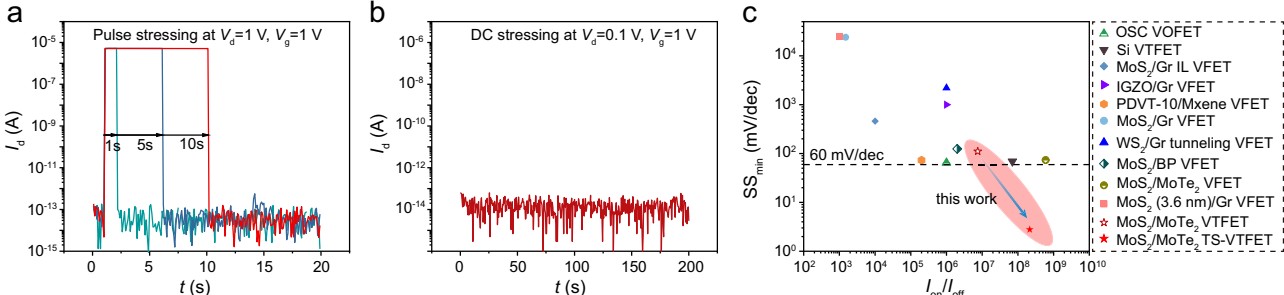

**Fig. 5 | Reliability and performance benchmark of the steep-slope TS-VTFET. a** DC pulse stressing test of the TS-VTFET. **b** DC stressing of the TS-VTFET under the threshold voltage of the TS cell. **c** Comparison of SS and $I_{on}/I_{off}$ ratio at room temperature between the demonstrated TS-VTFET and previously reported VTFETs, including the organic semiconductor (OSC)-based vertical organic FET (VOFET)[13], Si VTFET[9], graphene/MoS$_2$ VTFET[38], IGZO/graphene VTFET[14], VTFET with MXene Ti$_3$C$_2$T$_x$/Organic semiconductor PDVT-10[20], MoS$_2$/graphene VTFET[39], WS$_2$/graphene vertical tunnelling transistors[16], black phosphorous (BP)/MoS$_2$ VTFET[29], MoS$_2$/MoTe$_2$ VTFET[40] and MoS$_2$/graphene VTFET[15]. Here, the $I_{on}/I_{off}$ refers to the drain current modulation ratio between the maximum (ON) and minimum (OFF) current level during gate voltage sweeping. Device parameters of the TS-VTFET are taken from the transfer characteristics measured at $V_d = 0.3$ V.

steep-slope current switching and an excellent ON-OFF current ratio exceeding $1 \times 10^8$, the demonstrated MoS$_2$/MoTe$_2$ heterojunction-based TS-VTFET could thus outperform the reported VTFETs in terms of the sub-thermionic switching and low-power operation. Moreover, the vdW heterostructure-based TS-VTFET can be integrated seamlessly using the dry transfer method and TS medium deposition compatible with the BEOL process, which is beneficial for future low-energy electronic device manufacture[2].

## Discussion

In conclusion, we have reported vdW heterostructure-based VTFETs with a sub-5 nm transport channel length and steep-slope switching capability by exploiting the atomic thickness of 2D semiconductors and the abrupt current switching effect in TS cells. The demonstrated device features outstanding sub-thermionic SS down to ~2.7 mV/dec at room temperature, a competitive sub-60 mV/dec region over 6 decades of drain current, a high on-off current ratio and a low leakage current. These key performance metrics of the proposed TS-VTFETs have been shown to outperform previous vertical transistors. Furthermore, the demonstrated TS-VTFETs illustrate excellent reliability properties, which is critical for future practical applications. Our results provide a proof-of-concept steep-slope VTFET which could potentially be a front runner for next-generation ultra-scaled and low-power digital logic device technology.

## Methods

### Device fabrication

Flakes of MoS$_2$ and MoTe$_2$ (both purchased from HQ Graphene@) were mechanically exfoliated onto polydimethylsiloxane (PDMS, Gel-Pak@) supports using a low-residue blue type (Nitto, Japan). The 2D MoS$_2$ and MoTe$_2$ flakes of interest were then transferred onto the substrates. The MoS$_2$/MoTe$_2$ vdW heterostructures were vertically stacked using a location-precise transfer platform (Shanghai Onway Technology, China). Metal contacts to the 2D layers were formed through dry-transfer of the Au metal stripes. vdW material-based samples were annealed at 300 °C in vacuum for 2 h. The TS components were fabricated on the VTFETs using standard photolithography, metal sputtering and dielectric layer deposition methods. The Al$_2$O$_3$/HfO$_2$ dielectric gate stack was deposited on heavily doped $n$-type Si using atomic layer deposition (ALD). HfO$_2$ thin films were deposited through ALD at 200 °C with [(CH$_3$)$_2$N]$_4$Hf (TDMAHf) and H$_2$O as the Hf precursor and the oxygen source, respectively. To encapsulate the HfO$_2$ film, an Al$_2$O$_3$ layer was in situ deposited using Al(CH$_3$)$_3$ (TMA) and H$_2$O at 200 °C. Prior to the vdW heterostructure transfer, oxygen plasma was conducted to passivation the Al$_2$O$_3$ layer. TS dielectric layer deposition was conducted by a radio frequency magnetron sputtering

with a Ta$_2$O$_5$ ceramic target at room temperature. The oxygen-deficient TaO$_x$ and oxygen-rich TaO$_y$ layers were sputtered with a pure Ar of ~0.1 Pa and a mixture of Ar: O$_2$ (1: 2) at ~0.5 Pa, respectively. Ag electrodes were deposited in the same chamber by switching to the DC magnetron sputtering mode. A detailed vdW heterostructure preparation flow is schematically shown in Supplementary Note 1.

### Characterizations

For the Raman spectra, all the vdW heterostructures were measured using a WITec Alpha 300 R with a 532 nm laser under 1 mW light intensity. TEM-related measurements were carried out using a probe aberration-corrected JEOL JEM-ARM200F with an Oxford X-MaxN 100TLE spectrometer, which was operated at 200 kV. Specimens of the vdW heterostructures for the TEM tests were prepared by Focused Ion Beam (FIB) milling (FEI Helios NanoLab 600i). A FS-Pro semiconductor parameter analyzer with a minimum current accuracy of 30 fA equipped with a Lakeshore cryogenic probe station (set at 25 °C and ~10$^{-6}$ mbar) was adopted to perform all the electrical measurements. Thickness characterisation of 2D flakes was performed using a commercial atomic force microscopy (AFM) system (AIST-NT Smart SPM 1000) under an ambient atmosphere. Conductive platinum-coated tips (Mikromasch HQ: NSC35/Pt) with a force constant of ~5 N/m and a tip radius of less than 30 nm have been used for all imaging modes. To elucidate the operation mechanism of the VTFET, the band profiles were calculated by using the finite difference (FD) method to solve the 2-D Poisson's equation.

## Data availability

All data supporting this study and findings are presented in the article and the Supplementary Information file in graphic form. Source data are available from the corresponding authors upon request.

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

## Acknowledgements

We acknowledge support from the National Key R&D Program of China under Grant No. 2023YFB4402303 (Z.-D. L.), the National Natural Science Foundation of China under Grant No. 62025402 (G.H.), 62274128 (Z.-D.L.), 92264202 (Z.-D.L.), 62090033 (Y. Liu), 62293522 (G.H.), 92364204 (Y. Liu), 61971375 (W.C.), the Zhejiang Provincial Natural Science Foundation of China under Grant No. LDT23F04023F04 (Z.-D. L.), LDT23F04024F04 (G. H.), LR21F010003 (W. C.), the Fundamental Research Funds for the Central Universities under Grant No. QTZX23079 (Z.-D. L.), and the Key Research and Development Program of Ningbo City under No. 2023Z071 (Z.-D. L.). Z.-D.L. would like to thank Dr. C. Zhao with Analytic & Testing centre of NPU for the assistance of device fabrication. Z.-D.L. would like to thank Dr. S. Cheng of School of Microelectronics, Xi'an Jiaotong University for the assistance of TEM characterization. J.S. and D.Z. acknowledges support by the Australian Research Council through Discovery Grants and the ARC Centre of Excellence in Future Low Energy Electronics Technologies (FLEET).

## Author contributions

Z.-D. L. conceived the concept, designed the experiments, analysed the data and wrote the manuscript. Q. Y., Z.-D. L., Y. Li and D. T. fabricated the samples. Z.-D. L. and Q. Y. carried out the electrical measurement. H. D., W. C. and Z.-D. L. discussed and performed the model simulation of devices. D. Z. and J. S. conducted the AFM measurement. Z.-D. L., X. G., Y. Liu, Y. H. and G. H. supervised the project, and revised the manuscript with comments from all authors. All authors participated in the discussion and editing of the manuscript.

## Competing interests

The authors declare no competing interests.
