## [Peer Review File · Nature Communications]

Steep-Slope Vertical-Transport Transistors Built from sub-5 nm
Thin van der Waals HeterostructuresEditorial Note: Parts of this Peer Review File have been redacted as indicated to remove third-party material where no permission to publish could be obtained.

Reviewers' comments:

Reviewer #1 (Remarks to the Author):

This work reports a vertical field-effect transistor that combines a gate-controllable van der Waals heterojunction and a metal filamentary threshold switch (TS). Through the combination of resistance switching in TaOx and the gate-controlled current modulation in VTFET, the device exhibits low SS (21 mV/decade) and high current modulation ratio exceeding 1×10^5 .

However, the reviewer is more inclined to reject this paper. The major concerns from the reviewer are that, the device structure shows insufficient novelty when compared to previous works, and the superiority of the results is also insufficient.

1. The combination of resistance switching in TaOx and the gate-controlled current modulation in VTFET, however, is exactly the same as a well-known device structure of 1T1R (one transistor one memristor). For the purpose of steep-slope transistor, the device structure is similar to the 1S1R (one selector one memristor). The similar idea and device structure has been reported for many times in several years ago, such as the Nat Commun 11, 6207 (2020).

2. For the device performance, the DC curves of the VFET in Fig3/Fig4 are also simple and straightforward to directly combine a FET in-series with a TaOx Memristor. Besides, the steep-slope current switching ON-OFF ratio is only around $100 \sim 1000$, not 10^5 . The abrupt current is from 10pA to 10nA, which is too small for the practical application.

Reviewer #2 (Remarks to the Author):

This manuscript 'Steep-Slope Vertical-Transport Transistors Built from sub-5 nm thin van der Waals Heterostructures' by Qiyu Yang et al. reports steep-slope current change in transfer curve in MoS₂/MoTe₂ vertical transistor combined with Ag/TaOx/Ag threshold switch (TS) device. Their device exhibits subthreshold swing (SS) of 21 mV/dec over four decades, which lower than the thermionic limit of 60 mV/dec, and a current modulation ratio exceeding 1×10^5 at room temperature. The author's claim that their concept for steep-slope SS is the first demonstration of a vertical transistor. However, several vertical transistors have been developed that exhibit the steep-slope characteristic by using band-to-band tunneling between vertically stacked semiconductor-TMD materials, firstly demonstrated in Nature 526, 91-95 (2015) and best values obtained in Nature Electronics, 1-8 (2022). In Nature Electronics, 1-8 (2022), the SS value exhibits 6.4 mV decade⁻¹ over four decades of drain current. The devices also have a current on/off ratio of approximately 106 and an on-state current density of $0.3 \mu\text{A} \mu\text{m}^{-1}$ at a drain bias of -1 V , which values are higher than the current manuscript.

In comparison with the previous TS device (Nat Commun 11, 6207 (2020)) as well, the previous TS device exhibits lower SS (4.5 mV decade⁻¹ over five decades) and higher on/off ratio (107). The only advantage of the current device over the previous TS device is the vertical structure, however, advantages of vertical structure is not observed in the current device. The advantage of vertical transistor is high current density through entire overlapping area between top electrode and semiconductor vertical channel (Science 336, 1140-1143 (2012), Nature materials 12, 246-252 (2012), Nat Electron 4, 342–347 (2021)), while current flow in conventional planar transistors is confined to shallow semiconductor surface. In this study, despite the TS-VTFET being a vertical structure, its current level is comparable to that of a horizontal structure in other studies. Therefore, I cannot recommend the publication of this manuscript in Nature Communications.

Some other comments are listed below, which may helpful for publication in other journals.

- In Fig.2d, on the left panel, 1 V is displayed in blue and -1 V in red. However, on the right panel, 1 V is shown in red and -1 V in blue, which caused some confusion. I would like to standardize the color for each voltage.

- The band diagram illustration in Fig.2g and its description are unclear and difficult to understand. It would be helpful if you could provide additional information, such as indicating the change in barrier width and the direction in which the carrier moves. Additionally, it would be better to add a band diagram including the three operational states of TS.

- It appears that there is an error in the x-axis label of Supplementary Figure 3. V_{ds} should be modified to V_{gs}.

Reviewer #3 (Remarks to the Author):

The manuscript presents a study on steep-slope vertical-transport field-effect transistors (VTFETs) using a gate-controllable van der Waals heterojunction and a metal filamentary threshold switch (TS). The authors successfully demonstrate ultra-scaled vertical transport channels (<5 nm) with sub-thermionic turn-on characteristics. By combining the resistance switching of the TS and gate-controlled current modulation of the VTFET, the integrated TS-VTFETs exhibit efficient current switching behavior. However, there are certain revisions suggested by the reviewer regarding material selection, data representation, and labelling, which should be addressed to strengthen the manuscript's scientific rigor and clarity. Once these revisions are incorporated, the manuscript has the potential to make a significant contribution to the field of VTFETs.

1. The rationale for selecting MoTe₂ and MoS₂ as the materials is not clearly explained. (1) If the authors aim to create a p-n junction for controlling the resistance of the VTFET, it would be helpful to understand why they didn't use unipolar p-type materials like WS₂ instead of ambipolar materials. The reviewer would appreciate clarification on this point. (2) Additionally, if the authors intend to change the resistance of the VTFET to control metal filament formations in the TS cell, why don't they solely utilize MoS₂? By applying a backgate voltage, they could modulate the resistance of the MoS₂ layer by toggling the channel on and off, without the need for creating a p-n junction. Does this significantly affect the

mechanism of the TS cell? (3) In line 149, the authors mention “the Fermi-level pinning of n-type MoS₂.” Could you please explain the meaning of this statement? Are the authors suggesting that their MoS₂ materials are n-type due to the Fermi-level pinning effect? It is worth noting that Fermi-level pinning is typically observed at interfaces rather than within the materials themselves.

2. Upon examining the optical image (Fig. 2b) and schematic illustration (Fig. 3a), it is evident that there is a considerable length of “semiconducting” MoS₂ channel between the source electrode and heterostructure, relative to the size of the active device area. This raises concerns regarding the assertion of a purely vertical structure of the device, as lateral carrier transport appears to exist within this device structure. It is important to address the potential effects of lateral carrier transport and take them into consideration when evaluating the device’s performance.

3. In line 164, the authors assert that the deposition of an AlO₃ layer aims to suppress interface defects when in contact with 2D materials. (1) It would be beneficial to provide references or previous research that support this claim. Additionally, the reviewer wonders why the authors did not utilize h-BN to suppress interface scattering instead. (2) The Methods section lacks information regarding the fabrication of the AlO₃/HfO₂ substrate. It would be valuable to include details about the roughness of the substrate and the uniformity of the 3-nm AlO₃ film.

4. The thickness of 2D materials hold significant importance in VFET research, particularly in the case of 2D material-based devices. The schematic illustrations of the device featuring monolayer TMDs may potentially mislead readers. It is essential for the authors to either revise the figures to depict multilayers or provide explicit remarks on each figure clarifying the layer thicknesses. This will ensure accurate interpretation and understanding of the device structure by the readers.

5. In line 243, the authors state that both sides of the Ag electrodes result in excellent volatile bidirectional switching. In order to support this claim, it would be necessary for the authors to provide switching data from a one-sided Ag electrode TS cell, similar to the format shown in Fig. 3b. By including this additional data, the authors can provide a more comprehensive analysis and comparison, further supporting their statement regarding the bidirectional switching behavior.

6. In line 283, the authors claim that an average SS of 27.5 mV/dec “over four orders” of magnitude of the channel current. However, upon careful examination, it appears that the observed range is less than four orders of magnitude. I recommend verifying and correcting the numbers based on the available data to accurately reflect the magnitude of the channel current.

7. Fig. 3f lacks a sufficient number of data points to establish statistical significance, particularly for the TS-VTFET, which only displays two data points in the plot. It is essential for the authors to include additional data points to adequately populate the plot and ensure that the statistical analysis is robust.

8. The transfer curves in Fig. S3 are inaccurately labeled at the x-axis, which should read as V_g (gate voltage) instead of V_d (drain voltage). I recommend correcting the labels to accurately reflect the variable being represented in the plot.

9. In Fig. S8a, a significant reduction in hysteresis is observed at $V_d = 1$ V, which the reviewer acknowledges as an important improvement for practical transistors. (1) It would be helpful to understand what factors contribute to this improvement compared to the hysteresis observed at $V_d = 0.3$ V. (2) It appears that at higher values of V_d , the turn-on voltages are shifted towards negative V_g . In such cases, for V_d values higher than 1 V (e.g., $V_d = 1.5$ V), will the transfer curve follow the same pattern as the transfer curve of the VTFET shown in Fig. 3e? Clarification regarding the behavior of the transfer curve at higher V_d values would enhance the understanding of the device's characteristics.

General comment: This work reports a vertical field-effect transistor that combines a gate-controllable van der Waals heterojunction and a metal filamentary threshold switch (TS). Through the combination of resistance switching in TaO_x and the gate-controlled current modulation in VTFET, the device exhibits low SS (21 mV/decade) and high current modulation ratio exceeding 1×10^5 .

However, the reviewer is more inclined to reject this paper. The major concerns from the reviewer are that, the device structure shows insufficient novelty when compared to previous works, and the superiority of the results is also insufficient.

Response: We are extremely thankful for the reviewer's thoughtful and constructive feedback on our manuscript. We have carefully considered your comments and have made significant improvements to the paper based on your suggestions. We sincerely appreciate the time and effort you have invested in evaluating our work.

We understand from the reviewer's comments that the reviewer has reservations about the novelty and device performance superiority of our research. **To address these concerns, in the revised manuscript, we have clarified the very important aspect of novelty and conducted new measurements on similar devices to provide significantly improved device performance metrics along with important insights into the field of steep-slope logic transistor technology.** Our work would provide a novel perspective on the challenging problem regarding low-power operation and channel length downscaling of the specific vertical transistor technology (which has been identified as vital obstacles to the development of VTFETs, Nat. Electron. 4, 325, 2021; Nat. Electron. 4, 342, 2021), as also pointed out by the reviewer 3 "*the manuscript has the potential to make a significant contribution to the field of VTFETs*". Please find detailed answers in the following authors' response section.

In the following, we have provided a set of answers to all the comments, to clarify the raised concerns. Please note that in order to answer the Reviewers' questions and better demonstrate the performance superiority of our proof-of-concept TS-VTFET device, in the past half-year, we have fabricated new samples with the same VTFET device structure and optimized TaO_x-based TS cell, and conducted electrical measurements using a new advanced semiconductor parameter analyzer (FS-Pro from Primarius-tech)

with a superior minimum current accuracy down to 30 fA, please see Fig. R1. With the fact that a much better instrument measurement limit can be obtained using the new facilities, the measured results indicate significantly enhanced device performance metrics (see Table R1), allowing us to benchmark our device performance metrics fairly against earlier reports on steep-slope devices now. We believe that the revisions we have made, including extensive responses to reviewer comments and changes to the manuscript, have significantly strengthened the paper and increased its potential impact.

Once again, we thank the reviewer for the comments which help us improve the quality of our work. We hope that the reviewer will consider the revised manuscript favourably.

Fig. R1 In the revised manuscript, we have fabricated new vdW heterojunctions with exactly the same thicknesses as the previous samples. A FS-Pro semiconductor parameter analyzer with a minimum current accuracy of 30 fA equipped with a Lakeshore cryogenic probe station (set at 25 °C and $\sim 10^{-6}$ mbar) was adopted to perform all the electrical measurements for the revised manuscript. **Please note, with the fact that a much better instrument measurement limit can be obtained using the new facilities, the measured results indicated significantly enhanced device performance, allowing us to benchmark our device performance metrics fairly against earlier reports on steep-slope devices now.**

Table R1. Improvement of the device performance metrics in the revision with optimized devices and advanced measurement facilities.

	VTFET		TS-VTFET	
	Initial submission	Revised submission	Initial submission	Revised submission
V_d	0.3 V	0.3 V	0.3 V	0.3 V
V_g	(-1 V, 1 V) 35 mV/step	(-1 V, 1 V) 10 mV/step	(-1 V, 1 V) 35 mV/step	(-1 V, 1 V) 10 mV/step
I_{ON}	2.38×10^{-6} A	7.37×10^{-6} A	1.68×10^{-6} A	5.31×10^{-6} A
I_{ON}/I_{OFF}	1.43×10^5	7.47×10^6	3.36×10^5	1.77×10^8
I_{60}	NA	NA	6.91×10^{-9} A	1.1×10^{-7} A
Sub-60 SS_{avg} (I_{60}/I_{OFF} , decades of I_d)	NA	NA	3.17 dec	6.56 dec
SS_{min} (@ Fow.)	NA	NA	21 mV/dec	2.77 mV/dec
Note	Please note that the measured off current from the TS-VTFET exceeds the lower limit of our equipment's capabilities in the revision, consequently leading to the generation of random and negative current values. Therefore, we opt to utilize the lower limit of current detection (30 fA) for device performance metrics statistical calculations, irrespective of the recorded current values.			

Comment 1: The combination of resistance switching in TaOx and the gate-controlled current modulation in VTFET, however, is exactly the same as a well-known device structure of 1T1R (one transistor one memristor). For the purpose of steep-slope transistor, the device structure is similar to the 1S1R (one selector one memristor). The similar idea and device structure has been reported for many times in several years ago, such as the Nat Commun 11, 6207 (2020).

Response 1: We understand the reviewer's concerns and have made significant improvements to the manuscript based on the reviewer's comments. In the following, we would like to clarify the very important aspect of novelty.

Indeed, the reviewer is correct that a seemingly similar 1T1R device concept has been proposed for the **conventional lateral transport field-effect transistors (LTFETs)** to overcome the 60 mV/dec limit on SS at room temperature, which is itemed as phase-FET (Nat. Commun. 6, 7812, 2015; IEEE IEDM 34.6.1-34.6.4, 2016; Nat. Commun. 11, 6207, 2020). Such a device concept based on the idea that the resistive phase-transition component is connected to the drain or source terminal of the LTFET can induce an abrupt phase transition during gate voltage sweeping, thus leading to the sharp SS behaviour of the transistor. However, we respectfully disagree with the

reviewer's regard that this 1T1R-like steep-slope LTFET would weaken the novelty and importance of our work. Next, we would like to present factual clarifications that address your concerns regarding the novelty of the proposed device concept:

(I) Firstly, our work proposes a **novel technological co-integration of 2D vertical transport field-effect transistors (VTFETs) and resistive threshold-switching (TS) cell, enabling both area-efficient and sub-thermionic current switching device features**. Such a system has not been proposed or published in the literature previously. We emphasize that this device concept is completely different from the "1 filament TS cell-1 2D LTFET" device structure as the reviewer mentioned (Nat. Commun. 11, 6207, 2020) due to the following facts:

a. Our device concept is novel regarding a new "1TS cell-1 2D VTFET" device structure, which is completely different from the conventional planar/lateral transistor platform, being of high scientific importance as a novel performance booster for the specific vertical transistor technology. For our device concept, the 2D transistor is of a **unique vertical structure** in which the electrical transport of the device is solely determined by the MoS₂/MoTe₂ heterostructure channel. Note that although the fabricated device has a lateral transport channel of the MoS₂ layer, the charge transport in our VTFET is dominated by the vertical MoS₂/MoTe₂ *p-n* diode and has no appreciable contribution from the lateral semiconductor channel. Therefore, benefiting from the solely vertical-junction controlled transport properties in the VTFETs, such a vertical-structure device has a high potential to be further scaled in the lateral direction for advanced low-footprint vertical transistors (please see Supplementary note 6 for detailed discussion), presenting a pure **VERTICAL** device. This vertical transistor structure would be superior compared to conventional lateral transistor structure in terms of footprint, channel length scaling, leakage, electrostatic control, parasitic resistance/capacitance, etc. (see IBM and Samsung publication: IEEE IEDM pp. 26.1.1-26.1.4, doi: 10.1109/IEDM19574.2021.9720561, 2021). In fact, **this VTFET structure has been demonstrated to be advantageous over conventional lateral transistors in applications like logic circuits, RRAMs, thin film transistor (TFT)-driven LEDs, etc.** (Nat. Electron. 4, 914, 2021; IEEE IEDM pp. 26.1.1-26.1.4, 2021; Adv. Funct. Mater. 30, 1907113, 2020). For example, in a PCM cell consisting of a PCM element and a driving transistor (see Fig. R2), the predicted circuit design with the GAA VTFET would result in a cell size of 5.3F²; in stark contrast, the PCM cell with the conventional lateral MOSFET would lead to a cell size of 9F² (IEEE Trans. Electron Devices, 58, 664, 2011). **Hence, the VTFET would present an exciting alternative technology route for future logic transistors (see IBM report,**

<https://research.ibm.com/blog/vtfet-semiconductor-architecture>), and research of the steep-slope VTFET with low-power operation is rather of high scientific and research importance in this specific research field. In short, our device concept of a “1 TS cell-1 VTFET” structure is different from the conventional “1 filament TS cell-1 2D LTFET” structure and is of high technology importance.

Fig. R2 An example showing that the GAA VTFET (similar to our 2D TS-VTFET in this work (d)) would lead to a smaller footprint in PCM memory cell design. The GAA VTFET would result in a cell size of $5.3F^2$; in stark contrast, the PCM cell with the conventional lateral MOSFET would lead to a cell size of $9F^2$ (IEEE Trans. Electron Devices, 58, 664, 2011).

b. Our device concept is novel regarding a fundamentally different working principle by combining the TS abrupt current switching and gate-tunable vertical charge transport in a 2D vdW heterojunction. Besides the device structure difference, our device operation relies on a different working principle compared to the “1 filament TS cell-1 2D LTFET” device as the reviewer mentioned (Nat. Commun. 11, 6207, 2020). The reported device simply operates with the conventional MOSFET working physics. However, the $\text{MoS}_2/\text{MoTe}_2$ heterojunction-based VTFET works with a different mechanism. At $V_d > 0$ V, the proposed device operates as a conventional forward biased p - n diode, where the current increases with increasing the V_d and the V_g due to the exponential growth of majority carriers in the n -type MoS_2 . The device turns to the high resistance state under negative V_g due to the reason that the MoS_2 layer is modulated to the electron depletion state by gate control. The vertical transport is dominated by the diffusion of majority carriers, and most of the voltage drop occurs across the vertical p - n junction. Built upon the vertical transport properties of a VTFET,

we further con-integrated the TS switching ability into the hybrid device, which further enables the steep-slope switching of the overall device with SS largely below 60 mV/dec. **This unique working principle featuring the integration of vertical charge transport and gate-modulable threshold current switching (TS) has never been reported, and it promises the potential that our sub-thermionic device concept can tackle the power scaling problem of future vertical transistor technology.**

(II) **Our device concept is novel regarding a promising demonstration of a low-power vertical transistor with both sub-5 nm channel and sub-thermionic SS behaviour, which provides a new solution for ultra-scaled sub-thermionic devices in the specific research field of vertical transistor technology.** In terms of the specific vertical transport field-effect transistor (VTFET) technology, **downscaling of the channel length/thickness along with reduction of the subthreshold swing have been identified as vital obstacles to the development of VTFETs**, concerning the device performance parameters including saturation current, high frequencies, operation power consumption, etc. (Adv. Funct. Mater. 30, 1907113, 2020; Nat. Electron. 4, 325, 2021; Nat. Electron. 4, 342, 2021). **Previous VTFETs can hardly be downscaled to a sub-5 nm channel length with high-performance at the same time (see Table R2).** Among the very few reports on VTFETs with a channel length of less than 5 nm, the reported devices generally show moderate performance metrics with low on-off ratio, current density, high subthreshold swing (SS), etc. **Hence, this demonstrates that being able to fabricate a VTFET with a pure 2D/2D vdW heterostructure showing sub-5 nm channel length, as in this work, along with sub-60 SS behaviour and a new device working mechanism, is indeed novel (see Table R2).** It is worth noting, the integration of the TS cell with the VTFET is totally in a vertical direction which will not cause any current degradation even down to the nanometre scale, thus preserving the high device current density performance parameter. For example, the oxide-based TS devices can retain high current density down to 100 nm² size, see Nat. Mater. 16, 101, 2017. We also note that although our work is for realizing a new idea towards vdW heterostructure steep-slope VTFETs, the presented device already meets the current density requirement of some LED-driven TFT applications (current density > 600 A/cm²), please see Table R2. Therefore, the proposed 2D steep-slope TS-VTFET would be practical to raise a new VTFET device concept for energy- and area-efficient transistor technology.

Table R2. Benchmark of the device performance metrics with previous VTFETs

Junction	Channel thickness (nm)	I_{ON}/I_{OFF}	I_{ON} current density (A/cm ²)	V_d (V)	V_g (V)	Sub-60 SS	Ref.	comment
Gr/Si	Bulk Si	$\sim 1 \times 10^3$	0.4	0.3	(-3, 3)	NO	Science 336, 1140, 2012	
Au/MoS ₂ /Gr	36	1.5×10^3	2600	-0.5	(-60, 60)	NO	Nat. Mater. 12, 246, 2012	
	9	3.5	2100					
Ag/MoS ₂ /Gr (vdW metal)	3.6	$\sim 1 \times 10^3$	NA	0.1	(-60, 60)	NO	Nat. Electron. 4, 342, 2021	Same group, same fabrication method
	0.65	26	NA					
Ag/MoS ₂ /Gr (vdW metal)	5	1.85×10^3	~ 100	0.01	(-60, 60)	NO	Nano Lett. 23, 8303, 2023	
Pt/MoS ₂ /Gr (vdW metal)	4.55	4.83×10^5	~ 10					
Gr/WS ₂ /Gr	~ 9	$\sim 1 \times 10^6$	~ 200	0.2	(-30, 30)	NO	Nat. Nanotech. 8, 246, 2012	
Au/DPA/Gr (organic SC)	~ 126	1.8×10^5	~ 0.06	-5	(-80, 40)	NO	Adv. Mater. 30, 1803655, 2018	
Au/PDVT-10/MXene	76	2×10^5	~ 0.058	-30	(-20, 10)	NO	Nat. Commun. 13, 2898, 2022	
MoS ₂ /MoTe ₂	4.5	1.77×10^8	689	0.3	(-1, 1)	YES	This work	
TFT to drive AlGaAs- and InGaN μ LED			~ 10	NA			Nature 614, 81, 2023	ON current density needed to drive the LEDs
TFT to drive the OLED			~ 100	NA			Sci. Adv. 4, eaas8721, 2018	

(III) Lastly, **our optimized TS-VTFET device can achieve highly competitive individual performance metrics compared to other state-of-the-art steep-slope transistor device concepts.** In Table R3, we summarize the performance metrics of recently published high-performance steep-slope logic transistors, mainly from high-profile Nature series journals and industry-favoured conference proceedings including IEEE IEDM and VLSI. In terms of several critical performance metrics of logic transistors, the proposed device in this work shows excellent properties in comparison with other emerging device concepts including BTBT tunnel-FET (TFET), negative-capacitance (NC)-FET, NC-2D TFET, filamentary and (insulator-metal transition) IMT phase-FET, and Dirac-source injection FET. The two most important figure of merits of steep-slope devices would be the sub-60 mV/dec region and I_{60} , which determines the drain current span with SS below 60 mV/dec and the point where the current exhibits a transition from sub-60 to super-60 mV/dec (App. Phys. Lett. 102, 013510, 2013). Normally, **the IRDS mandated standard for a practical steep-slope device would require a sub-60 mV/dec region over at least 4 decades of drain current modulation and the I_{60} as high as possible.** As shown in Table R3, our device is among the very few devices that exhibit sub-60 mV/dec region over 4 or even 6 decades of drain current (from 10^{-14} A to 10^{-7} A). Plus, our TS-VTFET exhibits I_{60} over 1×10^7

A, which is even comparable to the threshold drain current amplitude of most 2D MOSFETs. Thus, compared to the previous steep-slope logic transistor device concepts mentioned by the reviewer (Nat. Commun. 11, 6207, 2020) and others, our device with encouraging performance metrics demonstrates a very promising solution for future high-performance steep-slope logic transistors, which is advantageous in terms of miniaturization, device performance, low power consumption and circuit design.

Table R3. Benchmark of the device performance metrics of emerging steep-slope transistor device concepts from high-profile Nature series journals and industry-favoured conference proceedings.

material	device concept	V_d (V)	I_{ON}/I_{OFF}	I_{60}	SS_{min} (mV/dec)	SS_{avg} over 4 decades (mV/dec)	SS_{avg} over 5 decades (mV/dec)	Sub-60 region I_{60}/I_{OFF} (decades of I_d)	Ref.	comment
P-Ge/MoS ₂	band-to-band Tunnel-FET	0.1	1.8×10^7	$\sim 1 \times 10^{-9}$ A	3.8	31.1	NA	~ 4	Nature 526, 91, 2015	
P-Si/InSe	band-to-band Tunnel-triode	-1	$\sim 1 \times 10^6$	$\sim 1 \times 10^{-8}$ A/ μ m	6.4	34	NA	~ 4	Nat. Electron. 5, 744, 2022	Same group. same device structure with different 2D semiconductors
P-Si/MoS ₂	heterojunction-triode	4	2×10^7	NA	> 60	NA	NA	NA	Nano Lett. 20, 2907, 2020	
SnSe ₂ /WSe ₂ On HZO	NC-2D TFET	0.3	$\sim 1 \times 10^6$	$\sim 5 \times 10^{-10}$ A	10	55	NA	~ 4	Nat. Electron. 6, 658, 2023	
Si-FinFET with HZO	NC-FinFET	0.1	$\sim 3 \times 10^6$	$\sim 1 \times 10^{-10}$ A	~ 58	NA	NA	~ 2	Nat. Electron. 6, 390, 2023	
MoS ₂ On HZO	NC-2D MOSFET	0.1	5×10^6	$\sim 2 \times 10^{-10}$ A	17.2	NA	NA	~ 3	IEEE IEDM 12.4.1-12.4.4, 2020	
Ag-HfO ₂ with MoS ₂ FET	Filament Phase-FET	0.2	5×10^6	$\sim 4 \times 10^{-8}$ A	2.5	3	4.5	~ 5	Nat. Commun. 11, 6207, 2020	
VO ₂ with Si MOFET	IMT Phase-FET	1.3	$\sim 1 \times 10^4$	$\sim 7 \times 10^{-5}$ A	8	NA	NA	~ 1.3	2016 IEEE VLSI doi: 10.1109/VLSIT.2016.7573445	
Graphene/MoS ₂	Dirac source FET	0.1	$\sim 1 \times 10^7$	$\sim 1.8 \times 10^{-5}$ A	29	NA	NA	~ 3	IEEE IEDM 12.5.1-12.5.4, 2020	
MoS ₂ /MoTe ₂ with TS cell	TS-VTFET	0.3	1.77×10^8	1.1×10^{-7} A	2.77	1.52	1.52	6.56	This work	

Overall, we believe that our work can be called novel as a new device concept (TS-VTFET) has been demonstrated for the first time, and there is a good device physics explanation of its uniqueness/benefits in its operation as sub-thermionic vertical transistors. Furthermore, we not only combined two cutting-edge technologies, i.e., TS cell and the 2D VTFET, to create a hybrid device that is clearly distinct from each individual component with completely new device physics, but we have also shown significant device performance metrics that are already suitable for some practical applications, e.g., the TFTs to drive μ LEDs. We understand that reviewer #1 had concerns that some seemingly similar device structures have been reported before (such as Nat. Commun. 11, 6207, 2020), as we argued above, one

may now find that our device is totally different in terms of device structure, device physics along with excellent performance metrics as steep-slope transistors. We strongly believe that the revised manuscript has significantly improved discussion on the novelty of our device along with the quality of the results. We hope that this detailed explanation clarifies the important aspects of the novelty of our work and that the constructive efforts we made can address the reviewer's concerns.

Action taken:

Following the above discussion, we have significantly modified the Introduction section, Supplementary information and any relevant part throughout the revised manuscript. We strongly believe that the revised version has significantly strengthened the novelty and the quality of the results. We hope that this detailed revision clarifies the important aspects of the novelty of our work and the constructive efforts we made can address the reviewer's concerns.

Comment 2: For the device performance, the DC curves of the VFET in Fig3/Fig4 are also simple and straightforward to directly combine a FET in-series with a TaOx Memristor. Besides, the steep-slope current switching ON-OFF ratio is only around 100~1000, not 10^5 . The abrupt current is from 10 pA to 10 nA, which is too small for the practical application.

Response 2: We thank the reviewer for raising this important point. However, we respectfully disagree with the reviewer's regards on that the VTFET electrical behaviour is simple and straightforward. In fact, we strongly believe that our device is novel in terms of the new device structure, new working physics and competitive device performance metrics in comparison with other emerging steep-slope transistor device concepts. Moreover, the proposed hybrid device structure can offer a viable solution to tackling the dilemma of performance degradation-size downscaling in the specific research field of vertical transistor technology. Please refer to Response 1 above for a detailed discussion on this.

Regarding the comment on the "ON-OFF ratio", we would like to clarify that the **term "current ON-OFF ratio" normally refers to the drain current modulation ratio between the maximum (ON) and minimum (OFF) current level during gate voltage sweeping**, rather than the steep-slope current switching ratio. The steep-slope current switching ratio is also an important figure-of-merit to describe a sub-thermionic transistor performance, which can be characterized as the current span in the sub-60 mV/dec region (I_{60}/I_{off}), where I_{60} is the point where the current exhibits a transition

from sub-60 to super-60 mV/dec (App. Phys. Lett. 102, 013510, 2013). **To address this concern, we have managed to show an improved TS-VTFET performance with the sub-60 mV/dec region over 6 decades of drain current (from 3×10^{-14} A to 1.1×10^{-7} A) along with a high I_{60} as of $\sim 1 \times 10^{-7}$ A.** In the meantime, the current ON-OFF ratio of the proposed device is as high as $\sim 1 \times 10^8$. **The overall device performance metrics of our TS-VTFET comfortably satisfy the IRDS mandated standard for a practical steep-slope device, i.e., a sub-60 mV/dec ratio (I_{60}/I_{off}) over at least 4 decades of drain current modulation and the I_{60} as high as possible.** A performance benchmark of our device with recently reported emerging steep-slope transistor device concepts from high-profile Nature series journals and industry-favoured conference proceedings is shown in Fig. R3 and Table R3. Obviously, the reviewer mentioned “abrupt current of 10 nA is too small” is not consistent with most of the reported I_{60} . **In fact, most of the state-of-art TFETs and NCFETs exhibit I_{60} below 10 nA, not to mention the optimized I_{60} of our device at ~ 100 nA level.** Therefore, we believe that the proposed TS-VTFET is competitive with other emerging sub-thermionic transistors in terms of crucial steep-slope device performance metrics including ON-OFF ratio, I_{60} and sub-60 mV/dec ratio, see Fig. R3 and Table R3.

Fig. R3 Benchmark of the critical sub-60 mV/dec figure-of-merits for the proposed TS-VTFET with other emerging steep-slope transistor device concepts. Data can also be found in Table R3.

Action taken:

Following the above discussion, we have significantly modified the manuscript and the SI. The benchmark of the proposed TS-VTFET device performance with other emerging steep-slope transistor device concepts has been added in the revised Supplementary note 14.

Part A-2: Responses to the comments from Reviewer #2

General comment: This manuscript ‘Steep-Slope Vertical-Transport Transistors Built from sub-5 nm thin van der Waals Heterostructures’ by Qiyu Yang et al. reports steep-slope current change in transfer curve in MoS₂/MoTe₂ vertical transistor combined with Ag/TaO_x/Ag threshold switch (TS) device. Their device exhibits subthreshold swing (SS) of 21 mV/dec over four decades, which lower than the thermionic limit of 60 mV/dec, and a current modulation ratio exceeding 1×10^5 at room temperature. The author's claim that their concept for steep-slope SS is the first demonstration of a vertical transistor.

Response: We are extremely thankful for the reviewer’s thoughtful and constructive feedback on our manuscript. We have carefully considered your comments and have made significant improvements to the paper based on your suggestions. We sincerely appreciate the time and effort you have invested in evaluating our work.

We understand from your comments that you have reservations about the novelty (previously reported BTBT TFETs) and crucial transistor device performance metrics (e.g., current density) of our research. **To address these concerns, in the revised manuscript, we have clarified the very important aspect of novelty of our work and conducted new measurements on similar devices to provide significantly improved device performance metrics along with important insights into the field of steep-slope logic transistor technology.** Our work would provide a **novel perspective on the challenging problem regarding low-power operation and channel length downscaling of the specific vertical transistor technology** (which has been identified as vital obstacles to the development of VTFETs, Nat. Electron. 4, 325, 2021; Nat. Electron. 4, 342, 2021), as also pointed out by the reviewer 3 “*the manuscript has the potential to make a significant contribution to the field of VTFETs*”. Please find detailed answers in the following authors’ response section.

In the following, we have provided a set of answers to all the comments, to clarify the raised concerns. Please note that in order to answer the Reviewers’ questions and better demonstrate the performance superiority of our proof-of-concept TS-VTFET device, we have fabricated new samples with the same VTFET device structure and optimized TaO_x-based TS cell, and conducted electrical measurements using a new advanced semiconductor parameter analyser (FS-Pro from Primarius-tech) with a superior

minimum current accuracy down to 30 fA, please see Fig. R1. With the fact that a much better instrument measurement limit can be obtained using the new facilities, **the measured results indicate significantly enhanced device performance metrics (see Table R1), allowing us to benchmark our device performance metrics fairly against earlier reports on steep-slope devices now.** We believe that the revisions we have made, including extensive responses to the reviewer's comments and changes to the manuscript, have significantly strengthened the paper and increased its impact.

Once again, we thank you for your comments which help us improve the quality of our work. We hope that you will consider the revised manuscript favourably.

Fig. R1 In the revised manuscript, we have fabricated new vdW heterojunctions with exactly the same thicknesses as the previous samples. A FS-Pro semiconductor parameter analyzer with a minimum current accuracy of 30 fA equipped with a Lakeshore cryogenic probe station (set at 25 °C and $\sim 10^{-6}$ mbar) was adopted to perform all the electrical measurements for the revised manuscript. **Please note, with the fact that a much better instrument measurement limit can be obtained using the new facilities, the measured results indicated significantly enhanced device performance, allowing us to benchmark our device performance metrics fairly against earlier reports on steep-slope devices now.**

Table R1. Improvement of the device performance metrics in the revision with optimized devices and advanced measurement facilities.

	VTFET		TS-VTFET	
	Initial submission	Revised submission	Initial submission	Revised submission
V_d	0.3 V	0.3 V	0.3 V	0.3 V
V_g	(-1 V, 1 V) 35 mV/step	(-1 V, 1 V) 10 mV/step	(-1 V, 1 V) 35 mV/step	(-1 V, 1 V) 10 mV/step
I_{ON}	2.38×10^{-6} A	7.37×10^{-6} A	1.68×10^{-6} A	5.31×10^{-6} A
I_{ON}/I_{OFF}	1.43×10^5	7.47×10^6	3.36×10^5	1.77×10^8
I_{60}	NA	NA	6.91×10^{-9} A	1.1×10^{-7} A
Sub-60 SS_{avg} (I_{60}/I_{OFF} , decades of I_d)	NA	NA	3.17 dec	6.56 dec
SS_{min} @ Fow.	NA	NA	21 mV/dec	2.77 mV/dec
Note	Please note that the measured off current from the TS-VTFET exceeds the lower limit of our equipment's capabilities in the revision, consequently leading to the generation of random and negative current values. Therefore, we opt to utilize the lower limit of current detection (30 fA) for device performance metrics statistical calculations, irrespective of the recorded current values.			

Comment 1: However, several vertical transistors have been developed that exhibit the steep-slope characteristic by using band-to-band tunneling between vertically stacked semiconductor-TMD materials, firstly demonstrated in Nature 526, 91-95 (2015) and best values obtained in Nature Electronics, 1-8 (2022). In Nature Electronics, 1-8 (2022), the SS value exhibits $6.4 \text{ mV decade}^{-1}$ over four decades of drain current. The devices also have a current on/off ratio of approximately 10^6 and an on-state current density of $0.3 \mu\text{A } \mu\text{m}^{-1}$ at a drain bias of -1 V , which values are higher than the current manuscript. In comparison with the previous TS device (Nat Commun 11, 6207 (2020)) as well, the previous TS device exhibits lower SS ($4.5 \text{ mV decade}^{-1}$ over five decades) and higher on/off ratio (107).

Response 1: We thank the reviewer for directly bringing up this important point. We understand the reviewer's concerns and have made significant improvements to the manuscript based on the reviewer's comments. **Aiming at offering a possible solution for future low-power and ultra-scaled device concept in the specific research field of vertical transistors, we believe that our work can be called novel as a new steep-slope device concept (2D heterojunction-based TS-VTFET) has been demonstrated for the first time, and there is a good device physics explanation of its uniqueness/benefits in its operation as sub-thermionic vertical transistors.**

Furthermore, we not only combined two cutting-edge technologies, i.e., TS cell and the 2D VTFET, to create a hybrid device that is clearly distinct from each individual component with completely new device physics, but we have also shown significant device performance metrics that are already suitable for some practical applications, e.g., the TFTs to drive μ LEDs. We strongly believe that the revised manuscript has significantly improved discussion on the novelty of our device along with the quality of the results. We hope that the following detailed explanation clarifies the important aspects of the novelty of our work and that the constructive efforts we made can address the reviewer's concerns. In the following, we would like to clarify the very important aspect of novelty to address the reviewer's concerns.

(I) **a.** As the reviewer rightly points out, two reported devices (P-Ge/MoS₂ and P-Si/InSe) sharing the same vertical BTBT TFET device concept have been reported (Nature 526, 91, 2015; Nat. Electron, 5, 744, 2022), showing a great example of steep-slope TFET. It is worth noting that the mentioned devices were not a 2D/2D vdW heterostructure but a kind of 2D/3D device; the scalability is thus not obvious as the quality of the interface may not be necessarily well controlled as in our 2D/2D interface case. Hence, despite the novelty at the respective time, we are not aware that the 2D/3D semiconductor heterojunction technology as the function part would be of enough potential to be adopted by the industry. In contrast, a pure 2D/2D vdW heterojunction would offer a precise and clean control of the interface quality and functionality with a high potential in terms of scalability (Science 353, 6298, 2016; Nat. Mater. 16, 170, 2017; Nature 499, 419, 2013). For sure, the debate would remain open to multiple solutions and their benchmarking. As for the comparison of important performance metrics of our steep-slope device, i.e., on-off ratio, I_{60} and SS, we have shown that our optimized TS-VTFET device in the revised manuscript can achieve highly competitive individual performance metrics even beyond the reports mentioned by the reviewer, please see Table R2, R3 and Fig. R3.

b. It is worth pointing out that the BTBT vertical TFET would **require a very strict selection of component semiconductors to fulfil the stringent requirement of the device working principle.** For example, as shown in Fig. R2, the P-Si/InSe heterojunction-based tunnel triode exhibited excellent sub-thermionic behaviour as mentioned by the reviewer (Nat. Electron, 5, 744, 2022); however, the P-Si/MoS₂ heterojunction-based triode from the same research group/authors only presented conventional transfer characteristic with SS >60 mV/dec (Nano Lett. 20, 2907, 2020). In contrast, the proposed co-integration of a TS cell with a VTFET would be a general

strategy even for other VTFETs, since this structure can always lead to steep-slope current switching properties as the abrupt current transition is merely due to the phase change of the TS cell. **This unique working principle featuring the integration of vertical charge transport and gate-modulable threshold current switching (TS) has never been reported, and it promises the potential that our sub-thermionic device concept can tackle the power scaling problem of future vertical transistor technology.** Moreover, MoS₂ and MoTe₂ share the same Mo element, so it would be more practical to fabricate MoTe₂/MoS₂ heterojunctions considering potential scalable production by only alternating the S and Te sources.

[REDACTED]

Fig. R2 The comparison of the device performance of P-Si/MoS₂ and P-Si/InSe 2D/3D heterojunction-based triodes. Results from the same research group, see Nat. Electron. 5, 744, 2022 and Nano Lett. 20, 2907, 2020.

(II) We would like to emphasize that the scope of our work is to offer a **novel perspective on the challenging problem regarding low-power operation and channel length downscaling of the specific 2D vdW heterostructure-based vertical transistor technology**, which has been identified as vital obstacles to the development of VTFETs (Nat. Electron. 4, 325, 2021; Nat. Electron. 4, 342, 2021). In this regard, **our device concept is advanced regarding a promising demonstration of a low-power vertical transistor with both sub-5 nm channel and sub-thermionic SS behaviour, which provides a new solution for ultra-scaled sub-thermionic devices in the specific research field of 2D heterostructure vertical transistor technology.** The channel length along with the SS behaviour of a vertical transistor would determine the device performance parameters including saturation current, high frequencies, operation power consumption, etc. (Adv. Funct. Mater. 30, 1907113, 2020; Nat. Electron. 4, 325, 2021; Nat. Electron. 4, 342, 2021). **Previous VTFETs can hardly be**

downscaled to a sub-5 nm channel length retaining high-performance at the same time (see Table R2). Among the very few reports on VTFETs with a channel length less than 5 nm, previously reported devices generally show moderate performance metrics with low on-off ratio, current density, high subthreshold swing (SS), etc. Hence, this demonstrates that being able to fabricate a VTFET with a pure 2D/2D vdW heterostructure of sub-5 nm channel length, as in this work, along with sub-60 mV/dec SS behaviour and a new device working mechanism, is indeed novel (see Table R2).

Table R2. Benchmark of the device performance metrics with previous VTFETs

Junction	Channel thickness (nm)	I_{ON}/I_{OFF}	I_{ON} current density (A/cm ²)	V_d (V)	V_g (V)	Sub-60 SS	Ref.	comment
Gr/Si	Bulk Si	$\sim 1 \times 10^3$	0.4	0.3	(-3, 3)	NO	Science 336, 1140, 2012	
Au/MoS ₂ /Gr	36	1.5×10^3	2600	-0.5	(-60, 60)	NO	Nat. Mater. 12, 246, 2012	
	9	3.5	2100					
Ag/MoS ₂ /Gr (vdW metal)	3.6	$\sim 1 \times 10^3$	NA	0.1	(-60, 60)	NO	Nat. Electron. 4, 342, 2021	Same group, same fabrication method
	0.65	26	NA					
Ag/MoS ₂ /Gr (vdW metal)	5	1.85×10^3	~ 100	0.01	(-60, 60)	NO	Nano Lett. 23, 8303, 2023	
Pt/MoS ₂ /Gr (vdW metal)	4.55	4.83×10^5	~ 10					
Gr/WS ₂ /Gr	~ 9	$\sim 1 \times 10^6$	~ 200	0.2	(-30, 30)	NO	Nat. Nanotech. 8, 246, 2012	
Au/DPA/Gr (organic SC)	~ 126	1.8×10^5	~ 0.06	-5	(-80, 40)	NO	Adv. Mater. 30, 1803655, 2018	
Au/PDVT-10/MXene	76	2×10^5	~ 0.058	-30	(-20, 10)	NO	Nat. Commun. 13, 2898, 2022	
MoS ₂ /MoTe ₂	4.5	1.77×10^8	689	0.3	(-1, 1)	YES	This work	
TFT to drive AlGaAs- and InGaN μ LED			~ 10	NA			Nature 614, 81, 2023	ON current density needed to drive the LEDs
TFT to drive the OLED			~ 100	NA			Sci. Adv. 4, eaas8721, 2018	

(III) Regarding the reviewer's concern over the performance metrics of our device with previously reported steep-slope devices, our optimized TS-VTFET device can achieve highly competitive individual performance metrics compared to other state-of-the-art steep-slope transistor device concepts. In Table R3 and Fig. R3, we summarize the performance metrics of recently published high-performance steep-slope logic transistors, mainly from high-profile Nature series journals and industry-favoured conference proceedings including IEEE IEDM and VLSI. In terms of the thorough evaluation of critical performance metrics of a logic transistor, the proposed device in this work shows excellent properties in comparison with other emerging

device concepts including BTBT tunnel-FET (TFET), negative-capacitance (NC)-FET, NC-2D TFET, filamentary and (insulator-metal transition) IMT phase-FET, and Dirac-source injection FET, please see Table R3 and Fig. R3. Two most important figure of merits of steep-slope devices would be the sub-60 mV/dec region and I_{60} , which determines the drain current span with SS below 60 mV/dec and the point where the current exhibits a transition from sub-60 to super-60 mV/dec (App. Phys. Lett. 102, 013510, 2013), respectively. Normally, **the IRDS mandated standard for a practical steep-slope device would require a sub-60 mV/dec region over at least 4 decades of drain current modulation and the I_{60} as high as possible**. As shown in Table R3, our device is among the very few devices that exhibit sub-60 mV/dec region over 4 and even 6 decades of drain current (from 10^{-14} A to 10^{-7} A). Plus, our TS-VTFET exhibits I_{60} over 1×10^7 A, which is even comparable to the threshold drain current amplitude of most 2D MOSFETs. **Thus, compared to the previous steep-slope logic transistor device concepts mentioned by the reviewer (e.g., Nat. Commun. 11, 6207, 2020)**, our device with encouraging performance metrics demonstrates a very promising solution for future high-performance steep-slope logic transistors, which is advantageous in terms of miniaturization, device performance, low power consumption and circuit design.

Action taken:

Following the above discussion, we have significantly modified the manuscript and the SI. We strongly believe that the revised version has significantly strengthened the novelty of our work and the quality of the results. We hope that this detailed revision clarifies the important aspects of the novelty of our work and the constructive efforts we made can address the reviewer's concerns. The benchmark of the proposed TS-VTFET device performance with other emerging steep-slope transistor device concepts has been added in the revised Supplementary note 14.

Table R3. Benchmark of the device performance metrics of emerging steep-slope transistor device concepts from high-profile Nature series journals and industry-favoured conference proceedings.

material	device concept	V_d (V)	I_{ON}/I_{OFF}	I_{60}	SS_{min} (mV/dec)	SS_{avg} over 4 decades (mV/dec)	SS_{avg} over 5 decades (mV/dec)	Sub-60 region I_{60}/I_{OFF} (decades of I_0)	Ref.	comment
P-Ge/MoS ₂	band-to-band Tunnel-FET	0.1	1.8×10^7	$\sim 1 \times 10^{-9}$ A	3.8	31.1	NA	~ 4	Nature 526, 91, 2015	
P-Si/InSe	band-to-band Tunnel-triode	-1	$\sim 1 \times 10^6$	$\sim 1 \times 10^{-8}$ A/ μm	6.4	34	NA	~ 4	Nat. Electron. 5, 744, 2022	Same group, same device structure with different 2D semiconductors
P-Si/MoS ₂	heterojunction-triode	4	2×10^7	NA	> 60	NA	NA	NA	Nano Lett. 20, 2907, 2020	
SnSe ₂ /WSe ₂ On HZO	NC-2D TFET	0.3	$\sim 1 \times 10^6$	$\sim 5 \times 10^{-10}$ A	10	55	NA	~ 4	Nat. Electron. 6, 658, 2023	
Si-FinFET with HZO	NC-FinFET	0.1	$\sim 3 \times 10^6$	$\sim 1 \times 10^{-10}$ A	~ 58	NA	NA	~ 2	Nat. Electron. 6, 390, 2023	
MoS ₂ On HZO	NC-2D MOSFET	0.1	5×10^6	$\sim 2 \times 10^{-10}$ A	17.2	NA	NA	~ 3	IEEE IEDM 12.4.1-12.4.4, 2020	
Ag-HfO ₂ with MoS ₂ FET	Filament Phase-FET	0.2	5×10^6	$\sim 4 \times 10^{-8}$ A	2.5	3	4.5	~ 5	Nat. Commun. 11, 6207, 2020	
VO ₂ with Si MOFET	IMT Phase-FET	1.3	$\sim 1 \times 10^4$	$\sim 7 \times 10^{-5}$ A	8	NA	NA	~ 1.3	2016 IEEE VLSI doi: 10.1109/VLSIT.2016.7573445	
Graphene/MoS ₂	Dirac source FET	0.1	$\sim 1 \times 10^7$	$\sim 1.8 \times 10^{-5}$ A	29	NA	NA	~ 3	IEEE IEDM 12.5.1-12.5.4, 2020	
MoS ₂ /MoTe ₂ with TS cell	TS-VTFET	0.3	1.77×10^8	1.1×10^{-7} A	2.77	1.52	1.52	6.56	This work	

Fig. R3 Benchmark of the proposed TS-VTFET device performance with other emerging steep-slope transistor device concepts. Data can also be found in Table R3.

Comment 2: The only advantage of the current device over the previous TS device is the vertical structure, however, advantages of vertical structure is not observed in the current device. The advantage of vertical transistor is high current density through entire overlapping area between top electrode and semiconductor vertical channel (Science 336, 1140-1143 (2012), Nature materials 12, 246-252 (2012), Nat Electron 4, 342-347 (2021)), while current flow in conventional planar transistors is confined to shallow semiconductor surface. In this study, despite the TS-VTFET being a vertical

structure, its current level is comparable to that of a horizontal structure in other studies. Therefore, I cannot recommend the publication of this manuscript in Nature Communications.

Response 2: We thank the reviewer for raising this important point. We totally agree with the reviewer that being of high current density is critical for applications based on VTFETs. Following the reviewer's comment, we have conducted a detailed survey of this performance parameter of previously reported vertical transistors mainly from high-profile journals including Science and Nature sister journals (including the references mentioned by the reviewer). **As shown in Table R2, it is obvious that the current density of our TS-VTFET (689 A/cm²) is fairly comparable to the best values of previously reported VTFETs.** Besides, we would like to emphasize that our optimized TS-VTFET device can further achieve highly competitive individual performance metrics compared to other state-of-the-art vertical transistor device concepts, including high on-off ratio, ultra-scaled channel thickness down to sub-5 nm, sub-thermionic SS behaviour, etc. (please see Table R2). Despite the fact that the results obtained from the TS-VTFET are encouraging, we believe that there is still room for further improvement. In fact, **as a proof-of-concept, the performance of 2D devices in a lab-made manner can vary a lot according to the fabrication procedure, material quality, etc.** Nevertheless, we note that **although our proof-of-concept work is for realizing a new idea towards vdW heterostructure steep-slope VTFETs, the presented device already meets the current density requirement of some LED-driven TFT applications (current density below 100 A/cm², see Nature 614, 81, 2023; Sci. Adv. 4, 8721, 2018), please see Table R2.** We also note that the integrating of an oxide-based TS cell vertically with the 2D/2D vertical junction through metal-oxide-metal deposition is technically favourable in terms of the CMOS process, and the oxide-based TS devices can retain high current density down to 100 nm² size, see Nat. Mater. 16, 101, 2017, which permits future area-scaling of our TS-VTFET. Therefore, we believe that the proposed vertical transistor device concept is competitive with other emerging vertical transistors in terms of the crucial device performance metric as of current density. Moreover, being with competitive individual performance metrics including high on-off ratio, ultra-scaled channel thickness down to sub-5 nm, sub-thermionic SS behaviour, etc., we trust that the proposed 2D steep-slope TS-VTFET with high current density would be practical to raise a new device concept for the specific VTFET technology, offering advanced solutions in energy- and area-efficient devices.

Action taken:

Following the above discussion, we have modified the manuscript to demonstrate the current density of the TS-VTFET. The benchmark of the proposed TS-VTFET device performance with other emerging vertical transistor device concepts has also been added in the revised Supplementary note 13.

Some other comments are listed below, which may helpful for publication in other journals.

Comment m1: In Fig.2d, on the left panel, 1 V is displayed in blue and -1 V in red. However, on the right panel, 1 V is shown in red and -1 V in blue, which caused some confusion. I would like to standardize the color for each voltage.

Response m1: We greatly thank the Reviewer for this suggestion. We have modified this figure to avoid possible confusion. Please see the Action taken below and the revised Fig. 2d.

Action taken:

We have changed the colour for Fig. 2d.

Fig. 2d Output curves at different gate voltage values for MoS₂ and MoTe₂ LTFETs, respectively.

Comment m2: The band diagram illustration in Fig.2g and its description are unclear and difficult to understand. It would be helpful if you could provide additional information, such as indicating the change in barrier width and the direction in which the carrier moves. Additionally, it would be better to add a band diagram including the three operational states of TS.

Response m2: We greatly thank the Reviewer for this suggestion. We have significantly modified Fig. 2g to provide additional information indicating the change in barrier width and the direction in which the carrier moves. We also revised Fig. 4c, which now

includes the band diagrams of the three operational states of TS in the hybrid TS-VTFET device. Relevant discussion with the new figures has also been added in the revised manuscript.

Action taken:

We have modified Fig.2g and 4c following the reviewer's suggestion. Relevant discussion was added in the revised manuscript accordingly.

Comment m3: It appears that there is an error in the x-axis label of Supplementary Figure 3. V_{ds} should be modified to V_{gs} .

Response m3: We greatly thank the Reviewer for carefully reading our manuscript. We have modified this figure as suggested, the revised figure can be found in Figure S3 of the revised SI.

Action taken:

Fig. S3 Transfer curves of MoS₂ (a) and MoTe₂ (b) LTFETs as well as the MoS₂/MoTe₂ vdW heterostructure-based VTFET (c). The device structure can be found in Fig. 2b in the manuscript.

Part A-3: Responses to the comments from Reviewer #3

General comment: The manuscript presents a study on steep-slope vertical-transport field-effect transistors (VTFETs) using a gate-controllable van der Waals heterojunction and a metal filamentary threshold switch (TS). The authors successfully demonstrate ultra-scaled vertical transport channels (<5 nm) with sub-thermionic turn-on characteristics. By combining the resistance switching of the TS and gate-controlled current modulation of the VTFET, the integrated TS-VTFETs exhibit efficient current switching behavior. However, there are certain revisions suggested by the reviewer regarding material selection, data representation, and labelling, which should be addressed to strengthen the manuscript's scientific rigor and clarity. Once these revisions are incorporated, the manuscript has the potential to make a significant contribution to the field of VTFETs.

Response: We are extremely thankful for the Reviewer's positive comments and constructive feedback on our manuscript. We have carefully considered your comments and have made significant improvements to the paper based on your suggestions. We sincerely appreciate the time and effort you have invested in evaluating our work.

In the following, we have provided a set of answers to all the comments, to clarify the raised concerns. **Please note that in order to answer the Reviewers' questions and better demonstrate the superiority of our TS-VTFET device performance, we have fabricated new samples with the same device structure and optimized TaO_x-based TS cell, and conducted electrical measurements using a new advanced semiconductor parameter analyzer (FS-Pro from Primarius-tech) with a superior minimum current accuracy down to 30 fA, please see Fig. R1. With the fact that a much better instrument measurement limit can be obtained using the new facilities, the measured results indicate significantly enhanced device performance metrics (see Table R1), allowing us to benchmark our device performance metrics fairly against earlier reports on steep-slope devices now.** We believe that the revisions we have made, including extensive responses to reviewer comments and changes to the manuscript, have significantly strengthened the paper and increased its impact.

Once again, we thank you for your comments which help us improve the quality of our work. We hope that you will consider the revised manuscript favourably.

Fig. R1 In the revised manuscript, we have fabricated new vdW heterojunctions with exactly the same thicknesses as the previous samples. A FS-Pro semiconductor parameter analyzer with a minimum current accuracy of 30 fA equipped with a Lakeshore cryogenic probe station (set at 25 °C and $\sim 10^{-6}$ mbar) was adopted to perform all the electrical measurements for the revised manuscript. Please note, with the fact that a much better instrument measurement limit can be obtained using the new facilities, the measured results indicated significantly enhanced device performance, allowing us to benchmark our device performance metrics fairly against earlier reports on steep-slope devices now.

Table R1. Improvement of the device performance metrics in the revision with optimized devices and advanced measurement facilities.

	VTFET		TS-VTFET	
	Initial submission	Revised submission	Initial submission	Revised submission
V_d	0.3 V	0.3 V	0.3 V	0.3 V
V_g	(-1 V, 1 V) 35 mV/step	(-1 V, 1 V) 10 mV/step	(-1 V, 1 V) 35 mV/step	(-1 V, 1 V) 10 mV/step
I_{ON}	2.38×10^{-6} A	7.37×10^{-6} A	1.68×10^{-6} A	5.31×10^{-6} A
I_{ON}/I_{OFF}	1.43×10^5	7.47×10^6	3.36×10^5	1.77×10^8
I_{60}	NA	NA	6.91×10^{-9} A	1.1×10^{-7} A
Sub-60 SS _{avg} (I_{60}/I_{OFF} , decades of I_d)	NA	NA	3.17 dec	6.56 dec
SS _{min} @ Fow.	NA	NA	21 mV/dec	2.77 mV/dec
Note	Please note that the measured off current from the TS-VTFET exceeds the lower limit of our equipment's capabilities in the revision, consequently leading to the generation of random and negative current values. Therefore, we opt to utilize the lower limit of current detection (30 fA) for device performance metrics statistical calculations, irrespective of the recorded current values.			

Comment 1: The rationale for selecting MoTe₂ and MoS₂ as the materials is not clearly explained.

(1) If the authors aim to create a p-n junction for controlling the resistance of the VTFET, it would be helpful to understand why they didn't use unipolar p-type materials like WS₂ instead of ambipolar materials. The reviewer would appreciate clarification on this point.

Response 1-1: We thank the Reviewer for raising this important point. Indeed, a rational pick of the constituent materials for functional devices would be vital. We agree with you that some unipolar *p*-type materials can also be applied to implement the VTFET by surveying the working mechanism of the device. For our VTFET, we normally use a positive V_d for the device operation. Briefly speaking, for $V_d > 0$, the VTFET works as a conventional forward-biased *p-n* junction. By applying a sufficiently large positive gate voltage, there is a significant current flow in the VTFET due to the diffusion of majority carriers from MoS₂ to MoTe₂ as shown in Fig. 2g. As V_g decreases, the current significantly decreases due to the exponential decrease of the majority carriers in MoS₂. **Thus, following the above physics picture, a unipolar *p*-type semiconductor in principle would serve the same purpose as the MoTe₂ does despite that the device performance would differ.** In this work, we chose MoTe₂/MoS₂ heterojunction as a model system to demonstrate the proposed TS-VTFET device concept for the following considerations: 1) As shown in Fig. R2, MoTe₂ is a *p*-terminal dominated ambipolar semiconductor with a relatively narrow bandgap (~1 eV for few-layer MoTe₂), and MoS₂ is a *n*-type semiconductor with a larger bandgap. The excellent gate-tunability of the band energies of MoTe₂ along with the reasonable energy band offset of the MoTe₂/MoS₂ heterojunction (Fig. R2b) would facilitate the voltage tunability of the junction electrical properties, which could lead to a great current on-off ratio (Science, 378, 296, 2022; Chem. Soc. Rev., 47, 3339, 2018). 2) Unlike the case of 2D *n*-type semiconductors, 2D *p*-type semiconductors with good properties are still scarce (Adv. Mater., 2206939, 2023). **Along with WSe₂, BP and Te, MoTe₂ are among the best few 2D *p*-type semiconductors with superior hole mobility and gate tunability (see Fig. R2c).** 3) MoS₂ and MoTe₂ share the same Mo element, it would be more practical to fabricate MoTe₂/MoS₂ heterojunctions considering potential scalable production, compared to the case of BP/MoS₂ or WSe₂/MoS₂ system, by only alternating the S and Te source.

[REDACTED]

Fig. R2 (a) Transfer properties of the MoTe₂ transistor showing *p*-terminal dominated ambipolar behaviour. (b) Schematic illustration of the band diagrams of the MoTe₂/MoS₂ heterojunction when separated. (c) A survey of reported 2D *p*-type semiconductors (Adv. Mater., 2206939, 2023).

Action taken:

Following the Reviewer' comment, we have added a brief discussion regarding the reason of selecting the MoTe₂/MoS₂ heterojunction as a model system in this work.

“As shown in Fig. 1c, the proposed VTFET exploits a vdW heterojunction by vertically-stacking the unipolar *n*-type (MoS₂) and the *p*-terminal dominated ambipolar (MoTe₂) semiconductors. Harnessing the merits including minimal interface trap states and large bandgap offset in high quality MoTe₂/MoS₂ vdW vertical heterostructures,²⁷ voltage-reconfigurable band alignment can be efficiently realised in such a heterojunction, giving rise to field-effect mediated transport behaviours.....Among the library of 2D semiconductor-based vdW *p-n* heterojunctions, such a MoS₂/MoTe₂ combination would be more practical to be fabricated from scalable production perspectives since only the alternation of S and Te source is needed for the same Mo source (unlike other 2D heterojunctions like MoS₂/WSe₂).³⁰”

*(2) Additionally, if the authors intend to change the resistance of the VTFET to control metal filament formations in the TS cell, why don't they solely utilize MoS₂? By applying a backgate voltage, they could modulate the resistance of the MoS₂ layer by toggling the channel on and off, without the need for creating a *p-n* junction. Does this significantly affect the mechanism of the TS cell?*

Response 1-2: We thank the Reviewer for raising this interesting point. The Reviewer's comment points out another type of device which consists of a TS cell and a conventional MoS₂ lateral transport field-effect transistor (LTFET), see Fig. R3. By modulating the resistance of the LTFET, the threshold switching of the TS component can for sure be efficiently controlled. Thus, in terms of the transistor conduction state-

controlled TS switching, the two device concepts share similar working physics and even electrical properties. **We next clarify that, however, the proposed TS-VTFET relying on the gate-controllable vertical p - n junction is different from the conventional lateral transistor and advantageous/novel in terms of the following aspects.**

Fig. R3 Schematic diagrams of the TS-LTFET and the TS-VTFET.

a. We first would like to emphasize again that we propose a new “1TS cell-1 2D VTFET” device structure in this work, which is completely different from the conventional planar/lateral transistor platform in terms of device structure and working physics, being of high scientific importance as a novel performance booster for the specific vertical transistor technology. For our device concept, the 2D transistor is of a **unique vertical structure** in which the device electrical transport is solely determined by the $\text{MoS}_2/\text{MoTe}_2$ heterostructure channel, see Fig. R3. Note that although the fabricated device has a lateral transport channel of the MoS_2 layer, **the charge transport in our VTFET is dominated by the vertical $\text{MoS}_2/\text{MoTe}_2$ p - n diode and has no appreciable contribution from the lateral semiconductor channel.** Therefore, benefiting from the **solely vertical-junction controlled transport properties in the VTFETs**, such a vertical-structure device has a high potential to be further scaled in the lateral direction for advanced low-footprint vertical transistors (please see Response 2 to the reviewer’s next comment), presenting a pure **VERTICAL** device.

In contrast, despite that a TS cell integration can drive the sub-60 mV/dec performance in the TS-LTFET structure as mentioned by the reviewer, such a device still suffers common challenges of a conventional lateral MOSFET technology, such as channel length scaling limitation, contact problem, etc. Therefore, albeit its potential in reducing the SS towards sub-60 mV/dec, such a combination of TS cell with LTFET would not be able to tackle some prominent problems of the conventional 2D MOSFETs, limiting its further development. This thus draws the importance of an alternative to the

b. The proposed vertical transistor structure would be superior compared to conventional lateral transistor structure in terms of footprint, channel length scaling, leakage, electrostatic control, parasitic resistance/capacitance, etc. (see IBM and Samsung publication: IEEE IEDM pp. 26.1.1-26.1.4, doi: 10.1109/IEDM19574.2021.9720561, 2021). In fact, **this VTFET structure has been demonstrated to be advantageous over conventional lateral transistors in applications like logic circuits, RRAMs, thin film transistor (TFT)-driven LEDs, etc. (Nat. Electron. 4, 914, 2021; IEEE IEDM pp. 26.1.1-26.1.4, 2021; Adv. Funct. Mater. 30, 1907113, 2020).** For example, **in the perspective of cell footprint, in a PCM cell consisting of a PCM element and a driving transistor (see Fig. R4), the predicted circuit design with the GAA VTFET would result in a cell size of $5.3F^2$. In stark contrast, the PCM cell with the conventional lateral MOSFET would lead to a cell size of $9F^2$ (IEEE Trans. Electron Devices, 58, 664, 2011). Hence, the VTFET would present an exciting alternative technology route for future logic transistors (see IBM report, <https://research.ibm.com/blog/vtfet-semiconductor-architecture>), and research of the steep-slope VTFET with low-power operation is rather of high scientific and research importance in this specific research field.** Our device concept of a “1 TS cell-1 VTFET” structure is different from the conventional “1 filament TS cell-1 2D LTFET” structure and is of high technology importance.

Fig. R4 An example showing that the GAA VTFET (similar to our 2D TS-VTFET in this work (d)) would lead to a smaller footprint in PCM memory cell design. The GAA VTFET would result in a cell size of $5.3F^2$; in stark contrast, the PCM cell with the conventional lateral MOSFET would lead to a cell size of $9F^2$ (IEEE Trans. Electron

(3) In line 149, the authors mention “the Fermi-level pinning of n-type MoS₂.” Could you please explain the meaning of this statement? Are the authors suggesting that their MoS₂ materials are n-type due to the Fermi-level pinning effect? It is worth noting that Fermi-level pinning is typically observed at interfaces rather than within the materials themselves.

Response 1-3: We thank the Reviewer for pointing this issue out. We agree with the Reviewer that Fermi-level pinning is typically observed at the semiconductor/metal interface. Here, by saying that the n-type MoS₂ is of Fermi-level pinning nature, we tried to hint that MoS₂ as a kind of 2D material with large amount of sulfur vacancy-induced dopant impurities typically shows an n-type nature. These sulfur vacancy-induced dopant impurities would keep the Fermi-level of MoS₂ always close to its conduction band edge even under strong gate voltage modulation, thus rendering MoS₂ a n-type 2D semiconductor (Science, 378, 296, 2022; ACS Nano, 11, 1588–1596, 2017). To avoid the confusion, we have modified the statement accordingly.

Action taken:

Following the Reviewer’s suggestion, we have modified the statement accordingly. “As shown in Fig. 1c, the proposed VTFET exploits a vdW heterojunction by vertically stacking the unipolar n-type (MoS₂) and the p-terminal dominated ambipolar (MoTe₂) semiconductors. Harnessing the merits including minimal interface trap states and large bandgap offset in high-quality MoTe₂/MoS₂ vdW vertical heterostructures,²⁷ voltage-reconfigurable band alignment can be efficiently realised in such a heterojunction, giving rise to field-effect mediated transport behaviours.”

Comment 2: Upon examining the optical image (Fig. 2b) and schematic illustration (Fig. 3a), it is evident that there is a considerable length of “semiconducting” MoS₂ channel between the source electrode and heterostructure, relative to the size of the active device area. This raises concerns regarding the assertion of a purely vertical structure of the device, as lateral carrier transport appears to exist within this device structure. It is important to address the potential effects of lateral carrier transport and take them into consideration when evaluating the device’s performance.

Response 2: We thank the Reviewer for this important comment. We agree with the Reviewer that the lateral semiconductor MoS₂ channel might raise concerns regarding the claim of a “pure” vertical-transport FET. It is obvious that the carrier transport path

in our VTFETs consists of the lateral and the vertical direction as marked in Fig. R5. An ideal VTFET would have a minimal lateral transport channel length in order to offer the ultra-scaled footprint in the 2D plane, albeit that the lateral conducting line at the bottom linking the vertical channel to the source electrode is unavoidable in VTFET device structure (*2015 Symposium on VLSI Technology (VLSI)*, 2015, pp. T26-T27). Unfortunately, it is impractical for us to make VTFET devices with lateral channel lengths smaller than $\sim\mu\text{m}$ scale due to the fabrication facility limitation in our group. To address the concerns over the effect of lateral transport on the electrical properties of our VTFETs, we instead carried out simulations to tackle this issue. **It is concluded that the vertical transport is the major factor determining the drain current in VTFETs as the junction resistance is higher than the lateral drift-diffusion resistance due to the presence of a p - n junction or a tunnelling barrier.**

Fig. R5 Schematic illustration of the vertical and lateral transport paths in the TS-VTFET

In the lateral direction, the length of MoS₂ is on the scale of a few microns, while in the vertical direction, the junction thickness is on the nanometer scale (~ 4.5 nm). The lateral transport in the VTFET proceeds by drift-diffusion with the intralayer carrier recombination within each material, while **the vertical transport is dominated by the diffusion of majority carriers or the band-to-band tunnelling** depending on the applied voltages. Fig. R6 and Fig. R7 show the simulated band diagrams in the lateral and vertical directions of the VTFET, respectively. At a forward bias V_d , both holes in MoTe₂ and electrons in MoS₂ are accumulated. **Most of the voltage drop occurs across the vertical p - n junction, and the band bending in the lateral transport direction within each semiconductor is very small as shown in Fig. R6 (a).** Consequently, the current in the VTFET is governed by the diffusion of majority carriers in the vertical p - n diode as shown in Fig. R7 (a), which indicates that **there is**

a very limited effect of the lateral channel on the overall VTFET transport properties. This result is in consistency with previous reports on MoS₂/WSe₂ *p-n* junctions (Nat. Nanotech., 9, 676, 2014; IEEE Trans. Electron Devices, 65, 4542, 2018). On the other hand, under reverse bias V_d , holes in MoTe₂ and electrons in MoS₂ are both depleted, respectively. The depletion in the junction region and quasi-Fermi level splitting at the reverse bias would result in large band bending in the lateral direction as depicted in Fig. R6 (b). Meanwhile, in the vertical direction, the electrons can tunnel quantum mechanically from MoTe₂ to MoS₂ according to the band profile of the diode under reverse bias V_d as shown in Fig. R7 (b). Such a band-to-band tunnelling is well controlled by the gate voltage and is also purely dominated by the MoTe₂/MoS₂ vertical junction. **The above analysis determines that the charge transport in our VTFET is dominated by the vertical MoS₂/MoTe₂ *p-n* diode and has no appreciable contribution from the lateral semiconductor channel.** This physics picture can be partially validated by the difference of the gate-controlled output properties between the MoS₂ LTFET and MoTe₂/MoS₂ VTFET (fabricated on the same flake, see Fig. R8). Clearly, under the negative gate voltage, the charge transport is dominated by the band-to-band tunnelling for the VTFET, showing a high current level under a negative drain voltage. In contrast, the MoS₂ LTFET shows no current running under this negative gate voltage, indicating a different charge transport mechanism. Therefore, benefiting from the solely vertical-junction controlled transport properties in the VTFETs, such a device has a high potential to be further scaled in the lateral direction for advanced low-footprint vertical transistors. As mentioned before, similar phenomena can be found in the atomically thin MoS₂/WSe₂ heterojunction junctions as shown in Fig. R9 (Nat. Nanotech., 9, 676, 2014) where the negligible effect of lateral transport on the atomically-thin *p-n* junction is obtained.

Fig. R6 Band diagrams in the lateral transport direction of the VTFET at different V_d under $V_g = 0 \text{ V}$. (a) $V_d = 0.3 \text{ V}$, and (b) $V_d = -0.3 \text{ V}$. The red (blue) dashed line represents the quasi-Fermi level of MoS₂ (MoTe₂). (c) The optical picture of the MoTe₂/MoS₂ VTFET device before capping a hBN layer.

Fig. R7 Band diagrams of the VTFET in the vertical transport direction at different V_d and $V_g = 0V$. (a) $V_d = 0.3V$, and (b) $V_d = -0.3V$. The dashed line represents the quasi-Fermi level.

Fig. R8 Output properties of MoTe₂/MoS₂ VTFET (a) and MoS₂ LTFET (b) under different gate voltages. (c) Optic photo showing the LTFET and VTFET devices fabricated with the same MoS₂ flake.

[REDACTED]

Fig. R9 Charge transport in an atomically thin $p-n$ heterojunction. (a) Schematic diagram of a vdW-stacked MoS₂/WSe₂ heterojunction with lateral metal contacts. (b) Band profiles in the lateral directions, and (c) band profiles in the vertical directions.

Action taken:

Following the Reviewer's comment, we have added the analysis with respect to the

lateral transport of the VTFET in the revised Supplementary note 6.

Comment 3: In line 164, the authors assert that the deposition of an AlO₃ layer aims to suppress interface defects when in contact with 2D materials.

(1) It would be beneficial to provide references or previous research that support this claim. Additionally, the reviewer wonders why the authors did not utilize h-BN to suppress interface scattering instead.

Response 3-1: We thank the Reviewer for this suggestion. Here, we choose a hybrid Al₂O₃/HfO₂ dielectric gate stack for two reasons. 1) As we wrote in the previous manuscript, the Al₂O₃ layer can be used to suppress the interface contacts when in contact with the 2D semiconductors. This claim can be supported by the paper published before (Adv. Mater., 27, 5230, 2015; Appl. Phys. Lett., 96, 142112, 2010). It is known that charge transport in atomically-thin 2D semiconductors, e.g. MoS₂, MoTe₂, etc., is largely influenced by extrinsic factors such as charge traps, defects and Coulomb impurities from the interface. As shown before (Adv. Mater., 27, 5230, 2015), the insertion of a thin layer of Al₂O₃ between monolayer WS₂ and SiO₂ can dramatically reduce the density of charge traps up by 49% and enhance the mobility by 2.3 times compared to the devices on bare SiO₂. Moreover, from our practical experience, the ALD-grown HfO₂ film always leads to large hysteresis for our 2D transistors due to the possible reason like interfacial charge trap formation. Such a situation can be much alleviated by adding a thin Al₂O₃ layer at the interface. 2) The Al₂O₃/HfO₂ dielectric gate stack was used by us due to the fact that the capping of Al₂O₃ on HfO₂ would reduce the leakage of the overall dielectric film. It is well known that the crystalline states formed in the HfO₂ thin film could lead to high leakage. Capping an amorphous Al₂O₃ layer is effective in inhibiting the crystallization of HfO₂ and thus preventing the leakage problem (Appl. Phys. Lett., 96, 142112, 2010; ACS Appl. Mater. Interfaces, 15, 16874–16881, 2023).

In this work, the reason we picked Al₂O₃/HfO₂ dielectric gate stack over the hBN is threefold: 1) In our experience, the Al₂O₃/HfO₂ gate stack is sufficient to offer great dielectric behaviour for 2D transistors. As can be seen from Fig. R10, the MoS₂ FET on Al₂O₃/HfO₂ gate stack shows excellent electrical properties with small hysteresis, proving that the interface charge trapping level is quite low. Such a high transistor performance is comparable to MoS₂/hBN transistor devices reported in the literatures. 2) Al₂O₃/HfO₂ dielectric gate stacks are conventional dielectric materials that are fully CMOS-compatible. Demonstration of our device using such kind of material would

hint at possible scalable production in the future. 3) Compared to hBN which has a dielectric constant of ~ 4 , Al_2O_3 and HfO_2 are both high- k dielectrics. Thus, using $\text{Al}_2\text{O}_3/\text{HfO}_2$ would effectively reduce the magnitude of gate voltage and enables low-power operation of the transistors.

Fig. R10 MoS_2 transistor on $\text{Al}_2\text{O}_3/\text{HfO}_2$ gate stack shows excellent electrical properties with small hysteresis

Action taken:

Following the Reviewer’s suggestion, we have revised the sentences about the purpose of capping Al_2O_3 layer on the HfO_2 thin film and added relevant references to support this claim.

“Note that the ultrathin Al_2O_3 layer is deposited on the HfO_2 for two purposes: 1) suppress the interface defects when in contact with the 2D layers;²⁷ 2) prevent the leakage favoured crystalline states formed in HfO_2 .²⁸”

(2) The Methods section lacks information regarding the fabrication of the $\text{AlO}_3/\text{HfO}_2$ substrate. It would be valuable to include details about the roughness of the substrate and the uniformity of the 3-nm AlO_3 film.

Response 3-2: We thank the Reviewer for the comment on the sample substrate. We have added the ALD fabrication process of the $\text{Al}_2\text{O}_3/\text{HfO}_2$ dielectric gate stack in the experimental section.

We have also provided the information regarding the surface of $\text{Al}_2\text{O}_3/\text{HfO}_2$ thin films. AFM topography scanning shows that the surface of the ALD-deposited Al_2O_3 film is of very low roughness which ensures the interface quality when contacting with the

adjacent 2D semiconductor layers.

Fig. R11 Topography and roughness of the ALD-deposited Al₂O₃ film

Action taken:

The substrate preparation process was added in the Methods section.

“The Al₂O₃/HfO₂ dielectric gate stack was deposited on heavily doped *n*-type Si using atomic layer deposition (ALD). HfO₂ thin films were deposited through ALD at 200 °C with [(CH₃)₂N]₄Hf (TDMAHf) and H₂O as the Hf precursor and the oxygen source, respectively. To encapsulate the HfO₂ film, an Al₂O₃ layer was in situ deposited using Al(CH₃)₃ (TMA) and H₂O at 200 °C. Prior to the vdW heterostructure transfer, oxygen plasma was conducted to passivation the Al₂O₃ layer.”

As suggested by the Reviewer, we have added the AFM results of the surface topography of Al₂O₃/HfO₂ thin films in the Supplementary note 10.

Comment 4: The thickness of 2D materials hold significant importance in VFET research, particularly in the case of 2D material-based devices. The schematic illustrations of the device featuring monolayer TMDs may potentially mislead readers. It is essential for the authors to either revise the figures to depict multilayers or provide explicit remarks on each figure clarifying the layer thicknesses. This will ensure accurate interpretation and understanding of the device structure by the readers.

Response 4: We thank the Reviewer for pointing out this issue. We have modified the figures accordingly following this comment to accurately show the readers the multilayer nature of the presented vdW heterostructures, please see Fig. R12 below. We have also stated the actual thickness of the fabricated vdW heterostructure devices in

the revised manuscript.

Fig. R12 Modified schematic illustration of vdW heterostructure-based devices.

Action taken:

The schematic illustrations of the vdW heterostructure-based devices have been modified throughout the revised manuscript, demonstrating its multilayer nature. The actual thickness of the MoS₂/MoTe₂ junction has also been clearly stated in the figure captions accordingly.

Comment 5: In line 243, the authors state that both sides of the Ag electrodes result in excellent volatile bidirectional switching. In order to support this claim, it would be necessary for the authors to provide switching data from a one-sided Ag electrode TS cell, similar to the format shown in Fig. 3b. By including this additional data, the authors can provide a more comprehensive analysis and comparison, further supporting their statement regarding the bidirectional switching behavior.

Response 5: We thank the Reviewer for this valuable comment. We have fabricated TS cells with both symmetric and asymmetric electrode configurations on the same substrate for comparison, i.e., Ag/TaO_x/TaO_y/TaO_x/Au and Ag/TaO_x/TaO_y/TaO_x/Ag as shown in Fig. R13. In stark contrast with the volatile bidirectional switching behaviour of Ag/TaO_x/TaO_y/TaO_x/Ag TS cells shown in the manuscript, the Ag/TaO_x/TaO_y/TaO_x/Au devices generally show unidirectional volatile switching properties. Such an Ag/TaO_x/TaO_y/TaO_x/Au cell can only go through a threshold current

switching when positive voltage is applied on the chemically active Ag side and remains insulating while negative voltage is applied on the rather inert Au electrode. This is because that a positive voltage applied on active Ag metal can easily trigger the growth of Ag-atom charged filaments, while a negative voltage bias on the inert Au fails to trigger the charge filament growth. Such a behaviour is commonly observed and studied in previous threshold switching selector works (Appl. Phys. Lett., 114, 193502, 2019; Adv. Electron. Mater., 8, 2101257, 2022), please refer to Fig. R14.

Fig. R13 (a)Optical image shown the fabricated Ag/TaO_x/TaO_y/TaO_x/Au and Ag/TaO_x/TaO_y/TaO_x/Ag cells sharing the same Ag/TaO_x/TaO_y/TaO_x structure. (b) Unidirectional threshold switching in an Ag/TaO_x/TaO_y/TaO_x/Au TS cell.

[REDACTED]

Fig. R14 Previous reports about the asymmetric electrode effect on the threshold switching properties.

Action taken:

A relevant discussion on the TS cells with asymmetric electrode configuration is provided in the revised Supplementary note 8.

“Supplementary note 8: The electrode configuration effect on electrical properties of the TS cell

Fig. S12 **a** Optical image showing the Ag/TaO_x/TaO_y/TaO_x/Au and Ag/TaO_x/TaO_y/TaO_x/Ag TS cells on the same substrate. **b** Typical unidirectional threshold resistive switching of the Ag/TaO_x/TaO_y/TaO_x/Au TS cell.

We have fabricated TS cells with both symmetric and asymmetric electrode configurations on the same substrate for comparison, i.e., Ag/TaO_x/TaO_y/TaO_x/Au and Ag/TaO_x/TaO_y/TaO_x/Ag as shown in Fig. S12. In stark contrast with the volatile bidirectional switching behaviour of Ag/TaO_x/TaO_y/TaO_x/Ag TS cells shown in the manuscript, the Ag/TaO_x/TaO_y/TaO_x/Au devices generally show unidirectional volatile switching properties. Such an Ag/TaO_x/TaO_y/TaO_x/Au cell can only go through a threshold current switching when positive voltage is applied on the chemically active Ag side and remains insulating while negative voltage is applied on the rather inert Au electrode. This is because a positive voltage applied on active Ag metal can easily trigger the growth of Ag-atom charged filaments, while a negative voltage bias on the inert Au fails to trigger the charge filament growth.^{7, 8}

Comment 6: In line 283, the authors claim that an average SS of 27.5 mV/dec “over four orders” of magnitude of the channel current. However, upon careful examination, it appears that the observed range is less than four orders of magnitude. I recommend verifying and correcting the numbers based on the available data to accurately reflect the magnitude of the channel current.

Response 6: We thank the Reviewer for pointing this issue out. In the previous measurements, the device off current is beyond the facility limit. Thus, we wrote that we anticipate that the sub-60 mV/dec current would span over 4 orders of magnitude. During the revision, we have managed to measure our new devices with a more advanced measurement facility. With the new results, we have clearly indicated the range of current changes in the revised figures. As can be clearly seen from Fig. 3e and

3f in the revised manuscript, the drain current abruptly jumps from the instrument measurement limit to $\sim 0.11 \mu\text{A}$ within a small gate voltage change of 10 mV, yielding an average SS of 1.52 mV/dec over 6 orders of magnitude of drain current spanning (from $\sim 10^{14}$ A to $\sim 10^{-7}$ A). This information can be found in the revised manuscript. Please note that the measured off current from the TS-VTFET exceeds the lower limit of our equipment's capabilities, consequently leading to the generation of random and negative current values. Therefore, we opt to utilize the lower limit of current detection (30 fA) for device performance metrics statistical calculations, irrespective of the recorded current values.

Fig. R15. Revised Figure 3e and 3f.

Action taken:

We have revised Figure 3e and 3f along with the related description in the manuscript. “Remarkably, in the forward gate bias sweep, the transfer curve of the TS-VTFET at $V_d = 0.3$ V shows excellent sub-threshold characteristics with an average SS of 1.52 mV/dec over seven orders of magnitude change in the channel current within the 10^{-14} A to 10^{-7} A range”.

Comment 7: Fig. 3f lacks a sufficient number of data points to establish statistical significance, particularly for the TS-VTFET, which only displays two data points in the plot. It is essential for the authors to include additional data points to adequately populate the plot and ensure that the statistical analysis is robust.

Response 7: We understand the Reviewer’s concern. The limited point SS value number of the TS-VTFET shown in Fig. 3f is due to the abrupt drain current switching over a small step of gate voltage change. As shown in Fig. Rx, the transfer curve of the TS-VTFET can be divided into three distinct regions, i.e., the steady off-current approaching the instrument measurement limit, the abrupt drain current switch and the gradually increased current after the TS-cell triggered threshold switching. By

calculating the SS points from the drain current with Matlab ($SS = dV_g/d\log(I_d)$), it is obvious that only the abrupt current switching region would yield a low SS value (please see Fig. R16). Considering that the drain current of the TS-VTFET jumps from $\sim 10^{-14}$ A to $\sim 10^{-7}$ A within a single 10 mV gate voltage step (please see Fig. R16b), it is mathematically plausible that only very limited SS points below 60 mV/dec can be obtained. Please note that the gate voltage sweeping step width was selected as small as 10 mV to obtain a steady current switching behaviour in the revised manuscript, however, gate voltage sweeping as large as 35 mV/step was needed to observe the steady current switching in the previous manuscript when using a different measurement facility. In contrast, the SS points of the VTFET in the same drain current scale would yield much more numbers due to its gradual current increase. Therefore, to better illustrate the steep-slope drain current switching properties, we showcase the important SS values (SS close to 60 mV/dec) in Fig. 3f (manuscript) for the TS-VTFET. **Please note that in contrast with the gradual SS change under 60 mV/dec as observed in NCFETs, TFETs, etc., the number of SS points that are below 60 mV/dec has been reported to be quite limited for phase-FETs with a very steep current switching over a small gate voltage range.** Such a phenomenon can be found in previous reports such as Nat. Commun. 2015, 6, 7812; Adv. Sci. 2021, 8, 2100208, please also refer to Fig. R17.

Fig. R16 (a) Typical transfer curves of the TS-VTFET and VTFET, respectively. (b) A zoomed-in view on the same plot of TS-VTFET transfer curve, showing three distinct regions. (c) SS point as a function of drain current.

[REDACTED]

Fig. R17 Typical transfer properties and point SS as a function of drain current for ultra-steep slope phase-FETs in previous reports.

Comment 8: The transfer curves in Fig. S3 are in accurately labeled at the x-axis, which should read as V_g (gate voltage) instead of V_d (drain voltage). I recommend correcting the labels to accurately reflect the variable being represented in the plot.

Response 8: We greatly thank the Reviewer for carefully reading our manuscript. We have modified this figure as suggested, the revised figure can be found in Figure S3 of the revised SI.

Action taken:

Following the Reviewer's suggestion, we have modified the figure accordingly.

Fig. S3 Transfer curves of MoS₂ (a) and MoTe₂ (b) LTFETs as well as the MoS₂/MoTe₂ vdW heterostructure-based VTFET (c). The device structure can be found in Fig. 2b in the manuscript.

Comment 9: In Fig. S8a, a significant reduction in hysteresis is observed at $V_D = 1$ V, which the reviewer acknowledges as an important improvement for practical transistors. (1) It would be helpful to understand what factors contribute to this improvement compared to the hysteresis observed at $V_D = 0.3$ V.

Response 9-1: We thank the Reviewer for bringing up this important suggestion. The physics involved in the TS-VTFET operation (including the hysteresis and the threshold voltage) would naturally be related to the resistive switching mechanisms of the TS devices, which is rather complicated from a quantitative perspective (Appl. Phys. Rev., 2, 031303, 2015) and slightly beyond the scope of our manuscript. To help the readers further understand the working principle and important factors affecting the operation of the proposed TS-VTFETs, below we provide a qualitative explanation of such. This can also be found in the revised SI.

The hysteresis in the transfer curve of TS-VTFET originates from the hysteretic current switching of the TS cell. It is clear that a hysteresis switching window exist in the IV curve of the TS cell whose width would be the voltage difference between the threshold switching and the hold point, i.e., $V_{\text{hyst-TS}} - V_{\text{th-TS}} - V_{\text{hold-TS}}$, see Fig. R18. In a TS-VTFET consisting of a TS cell and a VTFET, the two components share the supply voltage in a relation of $V_{\text{TS}} + V_D - V_D$ and $V_D / V_{\text{TS}} - R_{\text{VTFET}} / R_{\text{TS}}$ as shown in Fig. R18. Here, R_{VTFET} and R_{TS} can both be regarded as a variable resistor, whose resistance would be controlled by the V_g and the efficient voltage drop on the TS cell as V_{TS} , respectively. Therefore, in a hybrid system combining a TS cell and a VTFET, V_g actually controls the portion of voltage (V_D) across the TS cell through modulating the value of R_{VTFET} , which thus naturally translates the hysteresis switching of the TS cell ($V_{\text{th-TS}} - V_{\text{hold-TS}}$) to the hysteretic current switching in the transfer curve of the TS-VTFET ($V_{g\text{-th}} - V_{g\text{-hold}}$).

By choosing a V_D smaller than the V_{th} of the TS cell, even the maximum voltage drop on the TS cell (V_{TS}) cannot trigger the filament formation in the case that the VTFET is fully ON. Thus, the TS-VTFET keeps at an off state as the current level is totally controlled by the TS component no matter what the V_g is, see Fig. R18a. This unambiguously proves that the TS-cell operation is controlled by both V_D and V_g . Under a fixed V_g , the VTFET should remain a certain resistance state, therefore the TS-VTFET could be regarded as a variable resistor (TS cell) in series with a fixed resistor (VTFET at a fixed V_g). Assuming that the $R_{\text{VTFET}} / R_{\text{TS}}$ is almost at a constant value when applying a certain V_g on the VTFET, the efficient voltage drop on the TS component would get

higher with increasing the V_D . This indicates that it is possible to trigger the threshold switching of the TS cell with a lower V_g simply by increasing the V_D , that is, the transition from state 1 to state 2-3 occurs at a lower V_{g-th} (lower turn-on gate voltage for TS-VTFET). Thus, the V_{g-th} gets smaller with increasing the V_D . This is because that at a high V_D , larger V_{TS} can be obtained even if the VTFET is of a higher R_{VTFET} (lower V_g for n -type transistor), which is sufficient to trigger the threshold resistive switching of the TS component. This picture matches the observation of the relationship between V_d and V_{th} as shown in Fig. R18b. Please note that the current level of the TS-VTFET is totally controlled by the TS component at state 1. Since the hysteresis of the transfer curve (ΔV) is determined by both V_{g-th} and V_{g-hold} , we next address the relationship between V_{g-hold} and V_D . The reverse abrupt switching process of TS-VTFET is referred to as state 5 with gate voltage descending (see Fig. R18c), corresponding to the hold voltage of the TS cell. Similarly, a higher V_D would contribute to a larger V_{TS} , thus the V_{g-hold} of the TS-VTFET at higher V_D is likely to be smaller compared to the case that the TS-VTFET is under a smaller V_D . Indeed, we found the experiment results of the V_{g-hold} does follow the above pattern but show a small V_{g-hold} difference among the cases with various V_D . In fact, the V_{g-hold} shown in Fig R18a is quite close to each other for TS-VTFET under different V_D . This may be due to the complex rupture process of the filamentary TS cells, which involves the electric field effect, Joule heating, Gibbs-Thomson effect, etc. (Appl. Phys. Rev., 2, 031303, 2015; Adv. Sci, 10, 2301323, 2023).

Overall, based on the above physics picture, **the improvement of hysteresis in the TS-VTFET is strongly related to the applied V_D by considering the gate voltage controlled efficient voltage drop on the TS component.** Please note that a higher V_D can lead to a dramatic reduction of hysteresis but also of the I_{60} (current where the SS becomes 60 mV/dec), which thus requires a smart trade-off between each other during practical device operation. Following the working principles stated above, to overcome the hysteresis during TS-VTFET operation, it is important to reduce the hysteresis switching window of the TS cell (V_{th-TS} and $V_{hold-TS}$) itself, which thus needs a better understanding of the resistive switching in TS mediums and more advanced device engineering techniques.

Fig. R18 a Transfer curves of the TS-VTFET measured with different drain voltage. b Hysteresis voltage, I_{60} and V_{th} as a function of the applied drain voltage. Schematic demonstration of the origin of hysteresis during the TS-VTFET operation.

Action taken:

Following the Reviewer’s suggestion, we have added the above discussion into the revised SI. Please refer to “Supplementary note 11: Analysis of the electrical behaviours of the TS-VTFET during the current switching process” in the SI.

(2) It appears that at higher values of V_d , the turn-on voltages are shifted towards negative V_g . In such cases, for V_d values higher than 1 V (e.g., $V_d = 1.5$ V), will the transfer curve follow the same pattern as the transfer curve of the VTFET shown in Fig. 3e? Clarification regarding the behavior of the transfer curve at higher V_d values would enhance the understanding of the device’s characteristics.

Response 9-2: We thank the Reviewer for this important suggestion. Indeed, the turn-on voltage for TS-VTFET (V_{th}) decreases with increasing the V_d , please see Fig. R18a and 18c. We tried to further increase the V_d , however, this would cause a larger voltage across the TS cell as well. In our practical experience, the TS cells we made generally show sudden breakdown with applied voltage above 1 V considering its low threshold switching voltage. In the revised manuscript, we further measured the transfer curve of TS-VTFET at $V_d = 0.1$ V, 0.3 V, 0.6 V and 1 V, which clearly validate the pattern that the V_{th} gets smaller under higher V_d . A detailed explanation can be found in the Response 9-1 as shown above.

Cited from Response 9-1:

“By applying a V_D smaller than the V_{th} of the TS cell, even the maximum voltage drop on the TS cell (V_{TS}) cannot trigger the filament formation. Thus, the TS-VTFET keeps at an off state as the current level is totally controlled by the TS component no matter what the V_g is, see Fig. Rxa. This unambiguously proves that the TS-cell operation is controlled by both V_D and V_g . Under a fixed V_g , the VTFET should keep at a certain resistance state, the TS-VTFET could be regarded as a variable resistor (TS cell) in series with a fixed resistor. Assuming that the R_{VTFET}/R_{TS} is almost at a constant value when applying a certain V_g on the VTFET, the efficient voltage drop on the TS component would get higher with increasing the V_D . This indicates that it is possible to trigger the threshold switching of the TS cell with a lower V_g simply by increasing the V_D , that is, the transition from state 1 to state 2-3 occurs at a lower V_{g-th} (lower turn-on gate voltage for TS-VTFET). Thus, the V_{g-th} gets smaller with increasing the V_D . This is because that at a high V_D , larger V_{TS} can be obtained even if the VTFET is of higher R_{VTFET} (lower V_g for n-type transistor), which is sufficient to trigger the threshold resistive switching of the TS component. This picture matches the observation of the relationship between V_d and V_{th} as shown in Fig. Rxb.”

Action taken:

Following the Reviewer’s suggestion, we have added the above discussion into the revised SI. Please refer to “Supplementary note 11: Analysis of the electrical behaviours of the TS-VTFET during the current switching process” in the SI.

REVIEWER COMMENTS

Reviewer #1 (Remarks to the Author):

The authors claim they propose a novel technological co-integration of 2D vertical transport field-effect transistor (VTFET). However, the device structure is actually not vertical, at least the size of lateral part (~10 μ m) is much larger than the vertical part (~10nm). Then the authors claim the vertical part of heterojunction dominates the device performance (so that it can be called as vertical). This is true in view of the energy-band analysis. But the device performance mainly comes from the backgate voltage of V_{bg}. To remain the gate tunability, both backgate electrode and the lateral metal electrode are required.

That is to say, to get the novel vertical performance, we must keep the lateral device structure. Unless you can create side-wall gate-control somehow like a true 3D-transistor. Since it is not a true vertical device, the claimed advantageous over conventional lateral transistors will not solid, including the sub-5nm channel length, high integration scalability.

Next is the steep-slope performance. The device is exactly in a 1S1R-type, and the steep-slope comes from the abrupt switching of TS cell, which is commonly seen. If this can be called as a steep-slope transistor, then all of the 1S1R devices would say that they could achieve 10mV/dec with current modulation ratio exceeding 1×10^8 .

Reviewer #2 (Remarks to the Author):

In revised manuscript, the authors used a new facility that allows measuring the best properties of TS-VTFETs. Combining VTFET and TS, their TS-VTFET shows the highest performance compared to previous devices in on/off ratio, SS, and current density. Therefore, I recommend the publication of revised manuscript in Nature communications.

See the attached document for additional comments.

[Editorial Note: The comments in the document have been provided below]

Here are my comments for each concerns of reviewer #1.

The authors claim they propose a novel technological co-integration of 2D vertical transport field-effect transistor (VTFET). However, the device structure is actually not vertical, at least the size of lateral part (~10 μ m) is much larger than the vertical part (~10nm). Then the authors claim the vertical part of heterojunction dominates the device performance (so that it can be called as vertical). This is true in

view of the energy-band analysis. But the device performance mainly comes from the backgate voltage of V_{bg} . To remain the gate tunability, both backgate electrode and the lateral metal electrode are required. That is to say, to get the novel vertical performance, we must keep the lateral device structure. Unless you can create side-wall gate-control somehow like a true 3D transistor. Since it is not a true vertical device, the claimed advantageous over conventional lateral transistors will not solid, including the sub-5nm channel length, high integration scalability.

Comment: The vertical transistors is generally made of metal(graphene)- semiconductor-metal vertical stacks (NATURE MATERIALS VOL 12, 246, 2013, Nature Electronics volume 4, pages342–347 (2021)). Here the current is modulated by shifting the fermi-level of graphene electrode. In this case, current flows through the overall metal-semiconductor-metal overlapping area, resulting in very high current density. The device in this manuscript is closer to a vertical tunnel transistor using a semiconductor-semiconductor heterostructure (Nature volume 526, pages 91–95 (2015), Nature Electronics volume 5, pages744–751 (2022)). Vertical tunnel transistors have not only vertical heterojunction but also lateral TMD channel. Therefore, maximum current flow is limited in the lateral TMD channel.

I accepted this manuscript because on-current density (I_{on} in Table 3) is higher than previous MoS₂-TS FET (NATURE COMMUNICATIONS | (2020) 11:6207) by exchanging TMD lateral channel to TMD vertical tunnel junction, enabling high on/off ratio and low SS. However, As reviewer #1 comment, role of lateral part of MoS₂ channel is still arguable. Especially, the large gate modulation can occur in the MoS₂ channel (Fig. 2d, f), which is a critical point for turning the TS device on and off. This could become clear if the authors confine the gate electrode within the heterojunction region.

Next is the steep-slope performance. The device is exactly in a 1S1R-type, and the steep-slope comes from the abrupt switching of TS cell, which is commonly seen. If this can be called as a steep-slope transistor, then all of the 1S1R devices would say that they could achieve 10mV/dec with current modulation ratio exceeding 1×10^8 .

Comment: I had same concern as reviewer #1. However, it should have been covered in a previous paper (NATURE COMMUNICATIONS | (2020) 11:6207 | <https://doi.org/10.1038/s41467-020-20051-0>, which first introduced the steep-slope transistor based on abrupt switching of TS cell. Because previous paper was already accepted and published in Nature Communications, I didn't consider it in this review.

Reviewer #3 (Remarks to the Author):

I am grateful to the authors for their comprehensive response to my inquiries and their efforts to address the potential issues I raised about six months ago. Their explanations have partially resolved the questions I had previously. I am particularly impressed by their ability to optimize their devices, achieving device performance in terms of SS and On/Off ratio that surpasses what was previously reported in the literature, which was a significant weakness in this manuscript.

However, while the authors have made an effort to address the previous concerns about the device design through the application of simulation methods, it remains evident, as Referee #1 has rightly pointed out, that the current device structure is challenging to classify as either a vertical transistor or a sub-5nm channel length device.

To potentially resolve this issue, it may be beneficial for the authors to fabricate a device featuring a gate electrode solely beneath the heterostructure region.

One possible approach that comes to mind involves employing BN/graphite beneath the heterostructure and selectively etching away the surrounding material to ensure that the graphite backgate does not impact the lateral portion of the channel.

I believe that implementing this suggestion could help address the concerns.

Part A-1: Responses to the comments from Reviewer #1

Reviewer #1 (Remarks to the Author):

Comment 1: The authors claim they propose a novel technological co-integration of 2D vertical transport field-effect transistor (VTFET). However, the device structure is actually not vertical, at least the size of lateral part (~10 μ m) is much larger than the vertical part (~10nm). Then the authors claim the vertical part of heterojunction dominates the device performance (so that it can be called as vertical). This is true in view of the energy-band analysis. But the device performance mainly comes from the backgate voltage of V_{bg}. To remain the gate tunability, both backgate electrode and the lateral metal electrode are required. That is to say, to get the novel vertical performance, we must keep the lateral device structure. Unless you can create side-wall gate-control somehow like a true 3D-transistor. Since it is not a true vertical device, the claimed advantageous over conventional lateral transistors will not solid, including the sub-5nm channel length, high integration scalability.

Response 1: We would like to express our gratitude again for the reviewer's time to the peer-review process. We understand that you still have reservations over the vertical structure of our vdW-heterojunction VTFET. Regarding this, the reviewer #2 has provided a critical and constructive discussion, please see below input from reviewer #2 (marked in blue). Following reviewer #2's discussion, we have further provided a more detailed response to your comment, along with new experimental results on a VTFET with the lateral channel contribution almost eliminated.

“Comment from Reviewer #2: The vertical transistors is generally made of metal(graphene)-semiconductor-metal vertical stacks (NATURE MATERIALS VOL 12, 246, 2013, Nature Electronics volume 4, pages342–347 (2021)). Here the current is modulated by shifting the fermi-level of graphene electrode. In this case, current flows through the overall metal-semiconductor-metal overlapping area, resulting in very high current density. The device in this manuscript is closer to a vertical tunnel transistor using a semiconductor-semiconductor heterostructure (Nature volume 526, pages 91–95 (2015), Nature Electronics volume 5, pages744–751 (2022)). Vertical tunnel transistors have not only vertical heterojunction but also lateral TMD channel. Therefore, maximum current flow is limited in the lateral TMD channel. I accepted this manuscript because on-current density (I_{60} in Table 3) is higher than previous MoS₂-TS FET(NATURE COMMUNICATIONS | (2020) 11:6207) by exchanging

TMD lateral channel to TMD vertical tunnel junction, enabling high on/off ratio and low SS. However, as reviewer #1 comment, role of lateral part of MoS₂ channel is still arguable. Especially, the large gate modulation can occur in the MoS₂ channel (Fig. 2d,f), which is a critical point for turning the TS device on and off. This could become clear if the authors confine the gate electrode within the heterojunction region.”

1) First, we discuss that the proposed VTFET is a true vertical-transport dominated transistor. **Following the strategy proposed by Reviewer #2 and #3, we have fabricated a new MoTe₂/MoS₂ heterojunction VTFET with the junction area placed atop a confined back-gate electrode.** As can be seen from Fig. R1a, the drain electrode is fully confined within the bottom gate area so that the gate voltage can in principle merely modulate the electrical properties of the MoTe₂/MoS₂ heterojunction, not affecting the MoS₂ lateral channel. With increasing the back gate voltage (V_g) from -1 V to 1 V, the rectification behaviour of output $I-V_d$ curves gradually disappears, see Fig. R1b. The obtained electrical behaviour is the same as the case of the previous demonstrated MoTe₂/MoS₂ heterojunction VTFET (with a global gate electrode), see Fig. 2 of the main text. We have further measured the transfer characteristics of the fabricated VTFET (see Fig. R1c), which exhibits a clear gate-controlled OFF-to-ON conductance transition, just like the conventional MOSFET. It is obvious that the negative V_g can turn the entire device into the insulating state without affecting the MoS₂ lateral channel, thus verifying our statement that most of the V_d voltage drop occurs across the vertical $p-n$ junction. **Having analysed the above results, it would be safe to conclude that it is the MoTe₂/MoS₂ heterojunction which determines the transport properties of the device regardless the MoS₂ lateral channel, thus the proposed VTFET in this work is a true vertical-transport FET (VTFET).** Bearing this device physics in mind, we note that the MoS₂ lateral channel can be eliminated or largely scaled in future scalable manufacture, with the help of advanced nanofabrication techniques. A detailed discussion regarding the influence of lateral channel on device footprint can be found in the next answer. Nevertheless, as a proof-of-concept, we have unambiguously demonstrated that the MoTe₂/MoS₂ heterojunction VTFET is a pure vertical-transport transistor in this work, which can indeed be vertically scaled to a sub-5 nm transport channel length and be horizontally scaled to the vertical junction-limited $4F^2$ footprint (the same as the bulk material-based side-wall GAA 3D-VTFET), see Fig. R2.

Fig. R1 **a** Device structure of the MoTe₂/MoS₂ heterojunction VTFET with the junction area placed atop a confined back-gate electrode. The bottom gate along with the top drain electrode can only modulate the electrical properties of the overlapped MoTe₂/MoS₂ heterojunction. **b** Output curves at different gate voltage for MoTe₂/MoS₂ heterojunction VTFET. **c** Transfer curves of the VTFET measured with different V_d .

2) Regarding the Reviewer’s comment “*But the device performance mainly comes from the backgate voltage of V_{bg} . To remain the gate tunability, both backgate electrode and the lateral metal electrode are required. That is to say, to get the novel vertical performance, we must keep the lateral device structure. Unless you can create side-wall gate-control somehow like a true 3D-transistor.*”, we argue that, from an application perspective, this lateral part of the VTFET would NOT cause footprint increase in terms of the device structure. In fact, from a practical manufacture perspective, such as VTFETs formed in an array for area-efficient $4F^2$ DRAM or as a selector to the RRAM crossbar, lateral electrode parts connect to the source cannot be eliminated even in the side-wall gate-all-around (GAA) 3D vertical transistor (as the reviewer mentioned). In a side-wall GAA 3D vertical transistor, the bottom source contact will be connected using metal stripes to form the array, see Fig. R2. Likewise, for the case of 2D semiconductor heterojunction-based VTFET, the lateral channel can safely

remain without sacrificing the device area-efficiency, serving the purpose of conducting interconnection (covered with metal electrode) just like in the case of the side-wall GAA 3D vertical transistor, see Fig. R2c. Obviously, the two types of device structure won't result in footprint difference, see Fig. R2. We have already experimentally and theoretically verified that the lateral MoS₂ channel will not determine the overall transport behaviour of the TS-VTFET, please refer to Supplementary note 6, 15 and the above discussion. **This unambiguously demonstrates that the proposed TS-VTFET is a pure vertical transport-determined device, and our work would provide a proof-of concept vertical-type steep-slope device for future transistor technology.**

[REDACTED]

Fig. R2 Comparison of the side-wall GAA VTFET and the proposed TS-VTFET. **a** Side-wall GAA VFET (Materials Science in Semiconductor Processing, 2021, 134, 106046) for ultra-scaled 4F² footprint DRAM, IEEE IEDM 2023, 979-8-3503-2767-0/23 and IEEE IMW, 2023, pp. 1-4, doi: 10.1109/IMW56887.2023.10145977. **b** Side-wall GAA VFET as a selector for RRAM array, IEEE Trans. Electron Devices, 58, 664, 2011. **c** A proposed vdW heterojunction-based TS-VTFET with the lateral channel engineered to metallic phase, which thus would result in a further scaled vertical junction based VTFET.

Furthermore, to further improve the potential drain current density (especially for long lateral channel devices, note we did not see any current limitation problem for our fabricated few μm -long MoS₂ lateral channel devices in this work), we suppose that a channel doping/phase

engineering strategy can be used, especially for future scalable manufacture (for example, semiconductive 2H MoS₂ can be transformed to metallic conducting 1T MoS₂ using a chemical *n*-butyl lithium treatment method, see Nat. Mater. 2014, 13, 1128). In this way, conducting channel can be created between the MoS₂/MoTe₂ junction and the source contact, see Fig. R2c. However, **this proposed strategy requiring precise nano-fabrication technique (such as selective etching and layer-stop etching to expose the lateral MoS₂ channel only to chemical treatment) would be beyond the scope of the current work** (which mainly focuses on the proof-of-concept device structure/physics of TS-VTFET). We hope this strategy can be studied in future work, especially for the purpose of scalable integration of VTFET array.

Action taken:

Following the above discussion, we have added the new results obtained from the VTFET with confined gate electrode solely underneath the junction area into the Supplementary note 15.

Comment 2: Next is the steep-slope performance. The device is exactly in a 1S1R-type, and the steep-slope comes from the abrupt switching of TS cell, which is commonly seen. If this can be called as a steep-slope transistor, then all of the 1S1R devices would say that they could achieve 10 mV/dec with current modulation ratio exceeding 1×10^8 .

Response 2: We thank the reviewer for this comment. **However, we respectfully disagree with the reviewer's regard on the 1S1R device-based steep-slope transistor. The 1S1R-type transistor structure has been employed in previous planar Mott FET or phase-FET (Nature, 2023, 620, 501; IRDS 2022; Nat. Electron., 2018, 1, 442; Nat. Commun., 2020, 11, 6207), which has been widely explored as an innovative device concept for future logic transistor technology.** Following the reviewer #2's discussion (please see below the input from reviewer #2, marked in blue), we have further provided a detailed response to your comment.

“Comment from Reviewer #2: I had same concern as reviewer #1. However, it should have been covered in a previous paper (NATURE COMMUNICATIONS, 2020, 11:6207 /<https://doi.org/10.1038/s41467-020-20051-0>, which first introduced the steep-slope transistor based on abrupt switching of TS cell. Because previous paper was already accepted and published in Nature Communications, I didn't consider it in this review.”

[REDACTED]

Fig. R3 “1S1R” or “1 transistor-1 resistive change cell” device as an innovative steep-slope transistor strategy. **a** Phase-FET firstly proposed in *Nat. Commun.* 2015, 6, 7812, featuring a planar transistor with a phase change cell connected to its source. **b** Phase-FET along with TFET and NCFET are proposed as promising steep-slope logic switches, *Nat. Electron.* 2018,1, 442. **c** Mott-FET as an option for emerging logic devices, IRDS BC 2021. **d** Mott-FET with resistive change cell to the source of a transistor is listed as one of the future low-power transistor solutions, *Nature* 2023, 620, 501.

Transistors featuring a “1 transistor-1 resistive change cell” structure have long been regarded as a novel approach for future low power transistor technology, **which are termed as phase-FET** (*Nat. Electron.*, 2018, 1, 442; *Nat. Commun.*, 2020, 11, 6207) or **hyper-FET** (*Nat. Commun.*, 2015, 6, 7812), see Fig. R3a,b. Such a device concept employs a phase-change memory cell (Mott insulator, filamentary threshold switching cell, etc.) connected in series with the source of a transistor, to achieve an abrupt change of the drain current during the phase change induced memory-state switching (*Nature*, 2023, 620, 501). As a sub-class of the well-known Mott FETs (*Nature*, 2023, 620, 501; IRDS 2022), it has been widely explored as an innovative device concept for future logic transistor technology, see Fig. R3c,d. In a recent

perspective paper entitled “The Future Transistors” (*Nature*, 2023, 620, 501), this specific kind of Mott FET (phase-FET) with a phase-change cell to connect in series to the source of a transistor has been outlined as a promising device concept for logic transistor beyond MOSFET. Furthermore, phase-FET/Mott FET has also been identified by the industry as a practical option for emerging logic devices (IEEE IEDM 2016, 34.6.1-34.6.4, doi: 10.1109/IEDM.2016.7838542; IEEE VLSI 2016, pp. 1-2, doi: 10.1109/VLSIT.2016.7573445; The International Roadmap For Devices and Systems, IRDS 2021), see Fig. R3c. **In short, as an innovative steep-slope device concept, the “1S1R device” as termed by the Reviewer have been identified by both academia and industry as a promising and exciting approach for future logic transistor technology (Fig. R3), which deserves further exploration.**

There is no doubt that the phase-FET device concept is still in its infancy as it was first proposed in 2015 (*Nat. Commun.*, 2015, 6, 7812). The development of such a device technology would need significant research efforts to tackle the prominent challenges related to material selection, device structure, working physics, scalable integration, etc. **In this work, for the first time, we propose a vertical transport phase-FET, that is, the TS-VTFET. Our device concept is advantageous/innovative due to the following reasons:** 1) it presents new working physics/device structure completely different from the conventional planar phase-FET, which could **offer a solution to tackle the area-efficiency issue of planar phase-FET**; 2) the proposed device is a first-time demonstration of low-power vertical transistor with both sub-5 nm channel and sub-thermionic SS behaviour, which provides **a new solution for ultra-scaled sub-thermionic devices in the specific research field of vertical transistor technology**; 3) by combining the cutting technologies of phase-FET and VTFET as a performance booster, the optimized TS-VTFET device can achieve **highly competitive individual performance metrics compared to other state-of-the-art steep-slope transistor device concepts** (please refer to Supplementary Note 14 and Response to Reviewer #1 in the previous Response Letter).

Overall, we believe that the topic of Mott/phase-FET (featuring the 1S1R structure) as a steep-slope logic switch is an important and innovative topic for future transistors. The proposed TS-VTFET would improve the area-efficiency of the conventional planar phase-FET, and offer a viable solution to tackle the performance enhancement-channel downscaling dilemma for future low-power and area-efficient vertical transistor technology. Discussion related to the above statement has also been provided in the manuscript accordingly, please see below text copied from the manuscript (in purple).

“For 2D semiconductor-based LTFETs (Fig. 1a), the efficiency of the channel conductance modulation process is limited by the SS (minimum value of 60 mV/dec at room temperature)

due to the thermionic injection of carriers over an energy barrier.²² Notable innovative steep-slope device concepts have been established which exhibit reduced supply voltage and power consumption by improving the SS beyond the 60 mV/dec limit at room temperature, including but not limited to negative-capacitance FET,²¹ tunnelling-FET,⁶ Dirac-source FET,²³ impact ionization FET,²⁴ and phase-FET.^{25, 26} Especially, unlike other steep-slope transistor architectures, phase-FETs normally comprise disruptive components with different functionalities (that is, the two-terminal phase-transition component integrated in-series with the transistor), where the gate-voltage controlled transistor current transition along with the abrupt resistance switching of the phase-transition component would together lead to an overall sub-60 mV/dec steep-slope switching behaviours.²⁵⁻²⁸ Such a heterogeneous integration strategy of disruptive components can be feasibly extended to the conventional 2D semiconductor-based VTFETs (Fig. 1b). Therefore, integrating the two-terminal phase-transition device vertically with the 2D vdW heterostructure-based VTFET would naturally maximize the 3D construction capability of the device and result in a compact device architecture to further increase the transistor density on the chip, see Fig. 1c.”

Part A-2: Responses to the comments from Reviewer #2

Reviewer #2 (Remarks to the Author):

In revised manuscript, the authors used a new facility that allows measuring the best properties of TS-VTFETs. Combining VTFET and TS, their TS-VTFET shows the highest performance compared to previous devices in on/off ratio, SS, and current density. Therefore, I recommend the publication of revised manuscript in Nature communications.

Response: We appreciate the positive comments of the reviewer. We would like to express our gratitude again for the reviewer's contribution to the peer-review process, which has helped us to improve the quality of the manuscript significantly. Following the device strategy proposed by the Reviewer, we have fabricated a new MoTe₂/MoS₂ heterojunction VTFET with the back-gate electrode merely underneath the junction area. For the detailed discussion along with new results obtained from the fabricated device, please refer to Part A-1 Response 1 to Reviewer #1.

Part A-3: Responses to the comments from Reviewer #3

Reviewer #3 (Remarks to the Author):

Comment: I am grateful to the authors for their comprehensive response to my inquiries and their efforts to address the potential issues I raised about six months ago. Their explanations have partially resolved the questions I had previously. I am particularly impressed by their ability to optimize their devices, achieving device performance in terms of SS and On/Off ratio that surpasses what was previously reported in the literature, which was a significant weakness in this manuscript.

However, while the authors have made an effort to address the previous concerns about the device design through the application of simulation methods, it remains evident, as Referee #1 has rightly pointed out, that the current device structure is challenging to classify as either a vertical transistor or a sub-5nm channel length device.

To potentially resolve this issue, it may be beneficial for the authors to fabricate a device featuring a gate electrode solely beneath the heterostructure region. One possible approach that comes to mind involves employing BN/graphite beneath the heterostructure and selectively etching away the surrounding material to ensure that the graphite backgate does not impact the lateral portion of the channel. I believe that implementing this suggestion could help address the concerns.

Response: We would like to express our gratitude again for the reviewer's time and efforts to the peer-review process, which has helped us to improve the quality of the manuscript significantly. Following the device strategy proposed by the Reviewer, we have fabricated a new MoTe₂/MoS₂ heterojunction VTFET with the back-gate electrode merely underneath the junction area. For the detailed discussion along with new results obtained from the fabricated device, please refer to Part A-1 Response 1 to Reviewer #1.

REVIEWERS' COMMENTS

Reviewer #1 (Remarks to the Author):

I am grateful to the authors for their efforts on the additional devices and experimental data. Now I agree that the proposed VTFET is a true vertical-transport dominated transistor, in which the lateral part serves as the metal-interconnect function. However, I still have reservations to accept the paper, unless they could provide further evidence to show their vertical-type VTFET has surpassed the previous Nature-series publications. In fact, I didn't see much innovation/difference (on device structure) with previous works, and the great device performance also lacks statistical information.

The title "Steep-Slope Vertical-Transport Transistors Built from sub-5 nm Thin van der Waals Heterostructures" contains exactly the two points: VFET+small SS. But the local-gated VFET structure together with the steep-slope doesn't simultaneously exist in one device. Such additional data is necessary for both this paper and the whole VFET research field.

Reviewer #2 (Remarks to the Author):

It can now be published on Nature Communications.

Reviewer #3 (Remarks to the Author):

The author's latest revision addresses the problem regarding the impact of device design on its vertical characteristics. I recommend publishing this article.

Part C-1: Responses to the comments from Reviewer #1

Reviewer #1 (Remarks to the Author):

I am grateful to the authors for their efforts on the additional devices and experimental data. Now I agree that the proposed VTFET is a true vertical-transport dominated transistor, in which the lateral part serves as the metal-interconnect function. However, I still have reservations to accept the paper, unless they could provide further evidence to show their vertical-type VTFET has surpassed the previous Nature-series publications. In fact, I didn't see much innovation/difference (on device structure) with previous works, and the great device performance also lacks statistical information. The title "Steep-Slope Vertical-Transport Transistors Built from sub-5 nm Thin van der Waals Heterostructures" contains exactly the two points: VFET+small SS. But the local-gated VFET structure together with the steep-slope doesn't simultaneously exist in one device. Such additional data is necessary for both this paper and the whole VFET research field.

Response: We would like to thank you for your comments. We are glad that you agree that the presented VTFET is a true vertical-transport dominated transistor. Regarding your comment "...contains exactly the two points: VFET+small SS. But the local-gated VFET structure together with the steep-slope doesn't simultaneously exist in one device. Such additional data is necessary for both this paper and the whole VFET research field.", as we have already theoretically and experimentally demonstrated in previous response, the VTFET with lateral MoS₂ channel shares the same device physics with the local-gate VTFET structure. Furthermore, it has been shown that the both VTFETs exhibited similar device performance. Thus, both device structures can be selected for future practical applications of VTFETs without obvious device performance difference. As shown in the manuscript, we have thoroughly investigated the electrical properties of the TS-VTFETs with a vertical transport channel thinner than 5 nm, which showed sub-60 mV/dec SS over 6 decades of drain current. Thus, the two prominent features of the proposed device, i.e., steep-slope switching and sub-5 nm vertical-transport channel, have been showcased. As for the comment "*I still have reservations to accept the paper, unless they could provide further evidence to show their vertical-type VTFET has surpassed the previous Nature-series publications.*", as we have shown in previous response, the TS-VTFET device can achieve highly competitive individual performance metrics compared to other state-of-the-art steep-slope transistor device concepts. Moreover, as shown in the Supplementary Information Note 15, we have summarized the performance metrics of

recently published high-performance steep-slope logic transistors, mainly from **high-profile Nature series journals and industry-favoured conference proceedings including IEEE IEDM and VLSI**. In terms of several critical performance metrics of logic transistors, the proposed device in this work shows excellent properties in comparison with other emerging device concepts including BTBT tunnel-FET (TFET), negative-capacitance (NC)-FET, NC-2D TFET, filamentary and (insulator-metal transition) IMT phase-FET, and Dirac-source injection FET. Clearly, our device is among the very few devices that exhibit sub-60 mV/dec region over 4 or even 6 decades of drain current (from 10^{-14} A to 10^{-7} A). Plus, our TS-VTFET exhibits I_{60} over 1×10^7 A, which is even comparable to the threshold drain current amplitude of most 2D MOSFETs. Hence, compared to the previous steep-slope logic transistor device concepts, our device with encouraging performance metrics demonstrates a very promising solution for future high-performance steep-slope logic transistors.

Part C-2: Responses to the comments from Reviewer #2

Reviewer #2 (Remarks to the Author):

It can now be published on Nature Communications.

Response: We greatly appreciate your insightful comments during the review process and support for the publication of our manuscript in Nature Communications.

Part C-3: Responses to the comments from Reviewer #3

Reviewer #3 (Remarks to the Author):

Comment: The author's latest revision addresses the problem regarding the impact of device design on its vertical characteristics. I recommend publishing this article.

Response: We appreciate your precious comments throughout the review process of this work, which have helped us to improve the quality of the manuscript significantly.